# LLMInertia: Adaptive Counter-Inertial Reasoning to Improve Evidence Faithfulness in Large Language Models

**Xinxin You** [1]   **Xien Liu** [* 1]   **Chenwei Yan** [2]   **Siqi Song** [3]   **Chen Ning** [1]   **Kaiyin Zhou** [4 5]   **Shaohui Liu** [4 5]   **Ji Wu** [* 1 6 7]

## Abstract

Large Language Models (LLMs) frequently generate output that contradicts explicit input evidence, limiting their reliability in real-world applications. We identify **cognitive inertia** in LLMs—**a tendency to overly rely on co-occurrence associations learned during pretraining and to resist adaptation when conflicting input evidence appears**—as a critical factor behind such hallucinations. We further empirically show that adherence to input evidence declines as co-occurrence associations are strengthened—driven by either higher data frequency or intensified training. Inspired by human counter-inertial thinking, we propose an **adaptive counter-inertial reasoning** framework that probes input-related cognitive inertia in the LLM and generates adaptive counter-inertial reminders, which are then injected into the prompt to promote evidence-based reasoning. Experiments on co-occurrence induction datasets show that LLMInertia reduces hallucination rates by up to 35% and improves accuracy by up to 35.68%. Extensive evaluations on four context-rich summarization and QA datasets, across three LLM backbones of varying scales, further validate its effectiveness and robustness. Our work provides new insight into the causes of input-unfaithful hallucinations in LLMs, contributing to the development of more reliable AI.

[1]Department of Electronic Engineering, Tsinghua University, Beijing, China [2]School of Artificial Intelligence and Data Science, University of International Business and Economics, Beijing, China. [3]THiFly Health, Beijing, China [4]School of Computer Science (National Demonstrative Software School), Beijing University of Posts and Telecommunications [5]Key Laboratory of Trustworthy Distributed Computing and Service(BUPT), Ministry of Education [6]College of AI, Tsinghua University, Beijing, China [7]BNRist, Beijing, China. Correspondence to: Ji Wu <wuji_ee@mail.tsinghua.edu.cn>, Xien Liu <xeliu@mail.tsinghua.edu.cn>.

*Proceedings of the 43rd International Conference on Machine Learning*, Seoul, South Korea. PMLR 306, 2026. Copyright 2026 by the author(s).

## 1. Introduction

Large Language Models (LLMs) have achieved strong performance in a range of reasoning tasks, demonstrating the potential of artificial general intelligence (AGI) (Allen-Zhu & Li, 2024; Hendrycks et al.; Srivastava et al.). However, recent studies have revealed that LLMs exhibit unfaithfulness to input evidence—the factual or contextual (i.e., non-instructional) components of the input—by producing outputs that contradict information directly provided in the input (Zhang et al., 2023; Jiang et al., 2024). For example, as show in Figure 1, even when the prompt clearly states that the "examinations have ruled out Parkinson Disease", the LLM still overrides this explicit evidence. Such hallucinations present a significant obstacle to the practical adoption of LLM in real-world domains, where faithfully following input evidence (e.g., medical records in healthcare or case facts in law) is crucial (Srivastava et al.; Clark et al.; Lee et al., 2024). Violations can lead to poor reasoning, loss of user trust, reduced reliability of deployment, and potentially serious consequences (Liu et al., 2024; Zhu et al.).

Recent studies attribute input-evidence-unfaithful hallucinations to two primary factors. First, from the perspective of knowledge interaction, unfaithfulness frequently arises from knowledge conflicts, where dominant parametric priors contradict and overshadow provided contextual evidence (Xu et al., 2024b; Zhang et al., 2025), causing models to disregard explicit inputs due to confirmation bias (Xie et al., 2023). Second, from a modeling perspective, the causal language modeling objective encourages the exploitation of statistical regularities over deep semantic understanding (Kang & Choi, 2023; Liu et al., 2025; Zhang et al., 2024), aligning with the shortcut learning bias where models favor surface co-occurrences over demanding reasoning (Geirhos et al., 2020; Yuan et al., 2024; Zhang & Wu, 2024). However, while providing valuable insights, these works stop short of systematically quantifying specifically how the strength of these co-occurrence associations within parametric priors determines the severity of the model's resistance to explicit input evidence, underscoring the urgent need for dedicated investigation and mitigation.

In this work, we reveal that LLMs exhibit a human-like phenomenon of **cognitive inertia**, where models tend to overly

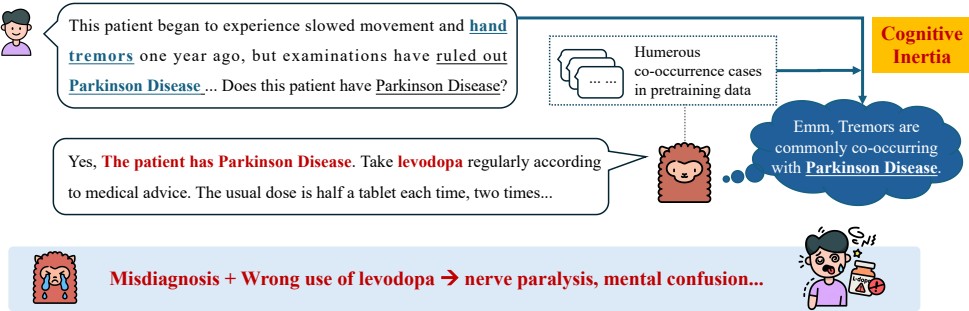

*Figure 1.* An example illustrating how LLMs generate medically incorrect outputs by over-relying on co-occurrence associations—such as between "hand tremors" and "Parkinson Disease"—while disregarding input evidence ("ruled out Parkinson Disease"). Such unfaithfulness to input evidence lead to misdiagnosis and inappropriate treatment, thereby presenting considerable risks to patient safety.

rely on co-occurrence associations learned during pretraining and resist adaptation when conflicting evidence appears in the input. We further show that *faithfulness declines as co-occurrence associations within parametric memory are strengthened—driven by either higher data frequency or intensified training.* Through controlled experiments on realistic medical and synthetic corpora that systematically manipulate both co-occurrence frequency and training intensity, we quantify the severity of input-evidence-unfaithful hallucinations under varying co-occurrence strengths. Experimental details are provided in Section 3, and results are shown in Figure 3.

Motivated by the above insights, we propose an **adaptive counter-inertial reasoning** framework, termed **LLMInertia**, to address the challenge of cognitive inertia in LLMs, as illustrated in Figure 2. Inspired by *human counter-inertial thinking—where individuals intuitively recognize counter-intuitive information and highlight critical cues to ensure faithful reasoning* (Saltiel & Woelfel, 1975; Turner & Sloutsky, 2024; Samadi et al., 2024)—our approach systematically probes the LLM to uncover high-frequency co-occurring associations relevant to the input. Based on these associations, LLMInertia automatically generate adaptive counter-inertial reminders and inject them into the prompt alongside the original input, explicitly guiding the model's attention to critical or easily overlooked input evidence. Experimental results on the co-occurrence induction datasets show that LLMInertia significantly reduces hallucination induction rates by up to 35% and improves accuracy by up to 35.68%. Extensive evaluations on four context-rich summarization and QA datasets, across three LLM backbones of varying scales, further validate its effectiveness and robustness. Our main contributions are as follows:

- We conduct the first systematic investigation into how excessive co-occurrence association within parametric priors trigger input-evidence-unfaithful hallucinations in LLMs, and quantify their influence on the model's resistance to input evidence. Building on these insights, we identify **cognitive inertia** as a central cause of such

faithful hallucinations.

- We present **LLMInertia**, an **adaptive counter-inertial reasoning** framework for LLMs. LLMInertia systematically probes the LLM to uncover input-relevant inertial associations, generates adaptive counter-inertial reminders, and injects these reminders into the prompt, thereby promoting faithful, evidence-based reasoning.

- Experimental results on the co-occurrence induction datasets show that LLMInertia significantly reduces hallucination induction rates by up to 35% and improves accuracy by up to 35.68%. Further evaluations across four summarization and QA benchmarks, spanning three LLM backbones, confirm its effectiveness and robustness.

## 2. Related Work

Factual faithfulness were studied in language generation—especially abstractive summarization—via graph-based fact modeling (Zhu et al., 2021), contrastive learning (e.g., CLIFF)(Cao & Wang, 2021), and multi-task/margin-based objectives to strengthen source grounding(Chen et al., 2022b). Building on this foundation, recent LLM studies focus on input-evidence-unfaithful hallucinations, which we summarize from two perspectives.

**Knowledge Conflict for Faithfulness Hallucination.** Recent research investigates the tension between LLMs' internal parametric knowledge and provided contextual evidence, termed *knowledge conflict* (Chen et al., 2022a; Xu et al., 2024b; Xie et al., 2024). While some studies suggest LLMs may suppress internal knowledge to align with context (Cheng et al., 2024), comprehensive investigations reveal that models frequently favor internal priors, either exhibiting strong confirmation bias against contradictory evidence (Xie et al., 2023) or allowing dominant parametric knowledge to obscure contextual information (Zhang et al., 2025). Despite these insights, the specific factors within parametric priors that trigger this unfaithfulness, and the quantitative extent of their influence on the model's resistance to input evidence, have not been investigated.

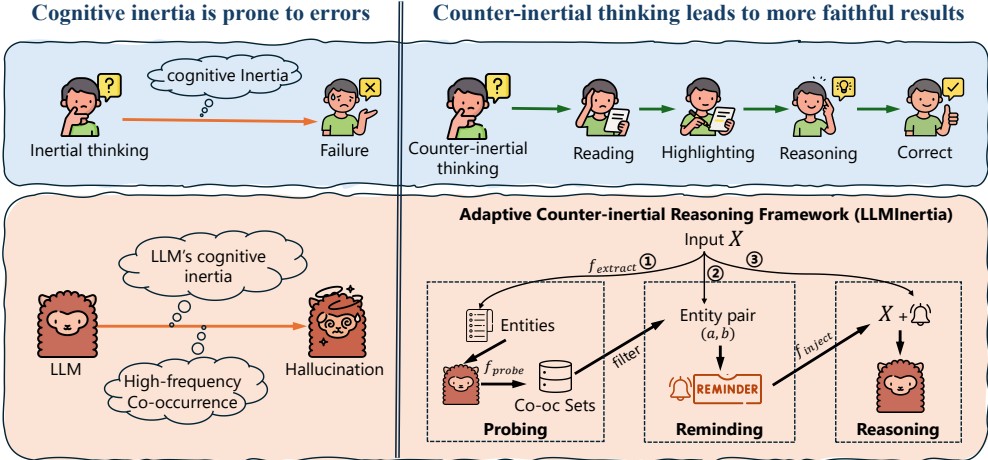

*Figure 2.* The left panel shows that cognitive inertia leads both humans and LLMs to overlook input evidence and rely mainly on high-frequency co-occurrence information, resulting in unfaithful reasoning. In contrast, the right panel illustrates that counter-inertial reasoning—actively identifying and emphasizing key evidence—leads to more accurate outputs. The lower panel shows the LLMInertia framework, which probes for input-relevant co-occurring associations, generates adaptive counter-inertial reminders, and injects them into the prompt to reduce hallucinations, thereby enhancing fidelity to input evidence.

**Shortcut Learning Bias for Faithfulness Hallucination.**
A parallel research has demonstrated that the objective of causal language modeling often drives next-token prediction by exploiting statistical regularities in the context rather than deeper semantic meaning (Liu et al., 2025; Zhang et al., 2024). This aligns with shortcut learning bias, where models favor easily learned surface co-occurrences over more demanding relational reasoning (Geirhos et al., 2020; Kang & Choi, 2023; Si et al., 2022; Du et al., 2023). In response, several studies have explored alignment-based fine-tuning and knowledge editing to improve output reliability (Yang et al., 2024b; Ju et al., 2024; Sun et al., 2024). However, the specific impact of co-occurrence learning association on input-evidence-unfaithful hallucinations remains insufficiently explored, highlighting the need—pursued in this study—to disentangle these mechanisms and design dedicated mitigation approaches.

## 3. Revealing Cognitive Inertia of LLMs

In this section, we present an experimental investigation into cognitive inertia in LLMs, with a particular focus on how over-learning co-occurrence associations contributes to input-evidence-unfaithful hallucinations. We design controlled experiments using specialized corpora for both real-world medical entities and synthetic pairs. After further pretraining on these datasets with systematically adjusted co-occurrence frequencies, we evaluate LLM responses to inputs containing conflicting information, enabling quantitative assessment of input-evidence-unfaithful hallucinations as co-occurrence strength varies. Details on data construction (Section 3.1), bias manipulation and continued pretraining (Section 3.2) and results analysis (Section 3.3)

### 3.1. Data Preparation and Corpus Construction

To investigate the effect of over-learning co-occurrence on the emergence and severity of input-evidence-unfaithful hallucinations, co-occurrence-induced training corpora and contradiction test sets are constructed.

**Induction Training Corpus.** To simulate real-world scenarios and isolate the co-occurrence bias inherent to the LLM backbone, we constructed 40 realistic medical entity pairs and 40 synthetic, fictitious pairs. The medical pairs were selected by statistically identifying high-frequency co-occurrences in the MIMIC dataset (Johnson et al., 2016). Each pair (e.g., "hand tremor", "Parkinson disease") was subsequently reviewed by medical professionals to confirm the absence of direct causal relationships. In contrast, the synthetic pairs (e.g., "blue map", "rubber elevator") were automatically generated via Deepseek-R1 (Guo et al., 2025) and deliberately constructed to lack any genuine semantic or causal associations. The full list of entity pairs is in Appendix A.1.

We used Deepseek-R1 to generate 100 diverse induction templates expressing co-occurrence. Each template was applied to all entity pairs, resulting in separate induction corpora of 4,000 samples each for medical and synthetic pairs. An example template is: "During the outpatient visit, the patient reported prominent symptoms of A, and the doctor recorded the presence of B; both are considered highly correlated clinical features." More templates are provided in Appendix A.2.

**Contradiction Test Corpus.** The test set to assess whether LLMs exhibit input-evidence-unfaithful hallucinations stemming from over-learned co-occurrence bias. Each sample

semantically expresses the premise "A is present, A and B are commonly associated, but B is absent." followed by the query "Is B present?" requiring a binary (yes/no) response. To enhance linguistic diversity, 25 distinct templates were created via Deepseek-R1 (Guo et al., 2025) to express this premise (e.g., "The main symptom is A, and B is not present, but A is recognized in some studies as possibly associated with B. Is B present?"). The full list of templates is in Appendix A.3. Applying these templates to each entity pair produced 1,000 medical and 1,000 synthetic test samples.

### 3.2. Bias Manipulation and Continued Pretraining

To systematically investigate the effects of co-occurrence bias, we conduct continued pretraining on three representative, open-source LLMs at different parameter scales: LLaMA-3-8B (Dubey et al., 2024), Qwen-2.5-7B (Yang et al., 2024a), and LLaMA-3-70B (Dubey et al., 2024). We perform full-parameter tuning for the 7B and 8B models, and parameter-efficient LoRA tuning for the 70B model (Hu et al.). Our goal is to systematically strengthen co-occurrence associations either by increasing their frequency in the training data or by intensifying the training process. To this end, we employ two strategies: (1) constructing 4000-example datasets with different co-occurrence data proportions (25%, 50%, 75%, 100%), supplemented with Wiki-en text (Guo et al., 2020) to ensure consistent size; and (2) fixing the co-occurrence ratio at 25% while varying the number of training epochs to induce different levels of co-occurrence exposure. Training hyperparameters and implementation details are provided in Appendix B.1.

### 3.3. Result and Analysis

We present detailed results showing how co-occurrence-driven cognitive inertia influences input-evidence-unfaithful hallucinations in LLMs, using two induction strategies and both medical and synthetic corpora settings.

**Evaluation Metrics.** We introduce the *hallucination induction rate* ($\eta$) to quantify the decline in LLM adherence to input information after co-occurrence induction:

$$\eta = \Delta N_{c \to i} / N_c \tag{1}$$

where $\Delta N_{c \to i}$ denotes the number of samples whose prediction switched from correct before induction to incorrect after induction, and $N_c$ is the total number of correct predictions before induction. We also report the *accuracy* (Obi, 2023) metric before and after induction.

**Cognitive Inertia Behavior 1: Excessive Co-occurrence Induces Faithfulness Hallucinations.** When we increase the proportion of co-occurrence data (Figure 3, unshaded columns), we observe a clear effect: the hallucination induction rate rises consistently as more co-occurrence-biased

samples are present during pretraining, and post-induction accuracy drops sharply across all models and benchmarks. Using LLaMA-3-8b under medical induction settings as an example, the accuracy decreases by **9.38%** to **37.70%** as the proportion of induction data increases, while the hallucination induction rate rises from **15.71%** to **44.56%**. This finding directly reveals that corpus imbalance leads models to overfit to high-frequency co-occurrence, thereby substantially amplifying superficial associations. Consequently, *the model becomes less faithful to explicit input evidence—particularly when such evidence contradicts biased associations—and this manifests as a clear increase in faithfulness hallucinations.*

**Cognitive Inertia Behavior 2: Extensive Pretraining Amplifies Faithfulness Hallucinations.** When the co-occurrence data ratio is fixed at 25% but the number of training epochs is increased (Figure 3, blue-shaded columns), we observe a similarly pronounced trend: hallucination induction rates generally show a steady increase, while post-induction accuracy consistently decreases as training progresses, despite minor fluctuations. This demonstrates that *cognitive inertia is a cumulative phenomenon—not only does dataset composition matter, but repeated exposure, even at small levels of bias, strongly entrenches co-occurrence associations within the LLM*. This phenomenon is especially pronounced in **domain-specific** settings, where data scarcity often leads practitioners to overfit on small datasets through extended training, thereby unintentionally increasing both the likelihood and severity of hallucinations.

**Medical vs. Synthetic Corpora: Isolating Pre-existing Bias.** Because we cannot verify whether the medical entity pairs appeared in the base LLM's pretraining, observed hallucinations may arise from independent induction data or from amplified pre-existing associations. In contrast, we constructed synthetic corpora with fictitious entity pairs that the model had never seen, ensuring that any hallucinations arose solely from our controlled induction. As a result, hallucination rates on synthetic data are lower and respond predictably to training interventions compared to medical corpora, which aligns with our suspicion that the foundation model was likely exposed to some medical entity pairs during pretraining. Collectively, these findings highlight that the tendency to persistently rely on pretraining co-occurrence associations—rather than adapt to contradictory input evidence—is a key factor contributing to unfaithful model outputs. Moreover, this phenomenon of cognitive inertia in LLMs is easily triggered under realistic imbalanced data distributions or over-training scenarios.

## 4. Mitigating LLM Unfaithfulness to Evidence

In response to cognitive inertia observed in LLMs, we introduce a novel **adaptive counter-inertial reasoning** frame-

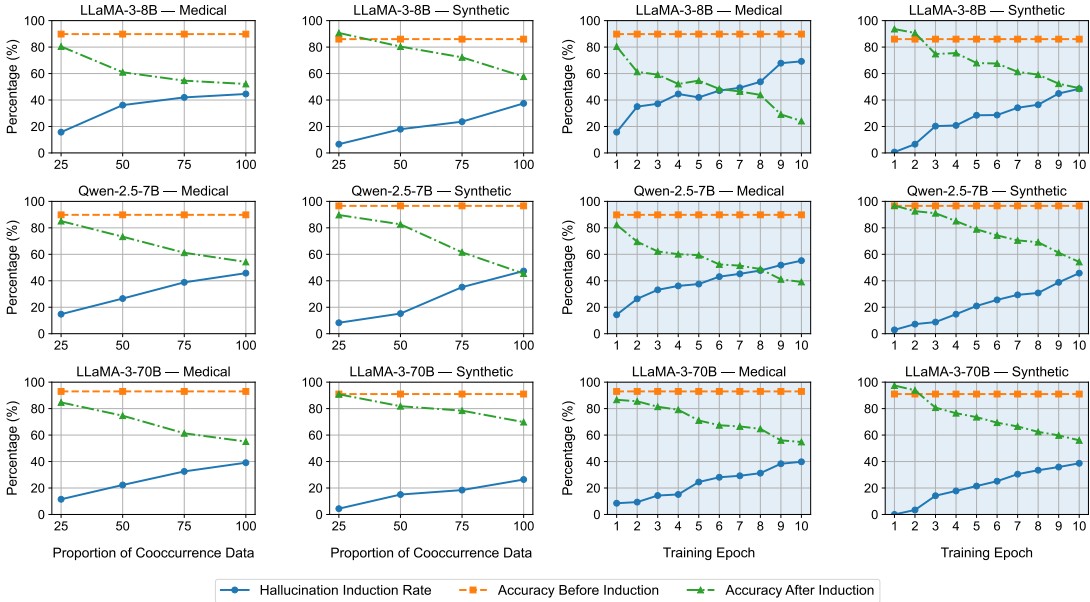

*Figure 3.* Impact of co-occurrence strength on input faithfulness across different LLMs on medical and synthetic datasets. The panels display performance trends under varying co-occurrence data proportions (unshaded columns) and training epochs (blue-shaded columns). As co-occurrence associations are strengthened (x-axis), the Hallucination Induction Rate (blue) rises while Accuracy After Induction (green) declines, indicating intensified cognitive inertia.

work—a simple yet effective approach that flexibly guides the LLMs' focus to potential input pitfalls, and enhances output faithfulness. The framework is illustrated in the lower right of Figure 2.

### 4.1. Adaptive Counter-Inertial Reasoning

**Cognitive Inertia Probe.** We design a probing process to uncover high-frequency internal associations in LLMs relevant to a given input. Specifically, given an input sequence $X = \{x_1, x_2, \ldots, x_n\}$ and an LLM $\mathcal{M}$, we first extract entities using $f_{\text{extract}}(X) = \{x_1, x_2, \ldots, x_m\}$, where $m$ is the number of extracted entities. The extraction function $f_{\text{extract}}$ is implemented via a tailored LLM prompt (see Appendix C.1.1), and achieves an F1 score above 87% on the base model (evaluation details in Appendix B.3), laying a solid foundation for subsequent processing.

For each extracted entity $x_i$, we apply a probing function $f_{\text{probe}}(\mathcal{M}, x_i)$ to identify its most frequent co-occurring entities within the LLM's internal knowledge. This yields an adaptive high-frequency co-occurrence association set:

$$\mathcal{K}_{\text{cooccur}} = \bigcup_{x_i \in f_{\text{extract}}(X)} \{(x_i, c) \mid c \in f_{\text{probe}}(\mathcal{M}, x_i)\} \quad (2)$$

where $f_{\text{probe}}(\mathcal{M}, x_i)$ returns the set of entities that most frequently co-occur with $x_i$ in model $\mathcal{M}$. The function $f_{\text{probe}}$ is implemented via a dedicated prompt (see Appendix C.1.2). The number of returned co-occurring entities per entity is a prompt-level hyperparameter, allowing us to effectively capture input-related cognitive inertia while limiting noise.

We use 3 for the induction corpus and 5 for QA and summarization tasks (see Section 4.8 for hyperparameter analysis).

**Counter-Inertial Reminders.** Based on the high-frequency co-occurrence set $\mathcal{K}_{\text{cooccur}}$, we introduce a mechanism for the automatic generation of *adaptive counter-inertial reminders*. For each entity pair $(a, b) \in \mathcal{K}_{\text{cooccur}}$, we check whether both $a$ and $b$ appear in the current input $X$. If this condition is satisfied, we generate a reminder message $r_{a,b}$ using the following adaptive template:

> **Reminder:** $a$ and $b$ may not necessarily be related. Please examine the original text carefully and make a thorough judgment.

The set of all reminders for input $X$ is thus defined as:

$$\mathcal{R}_{\text{AIR}} = \{r_{a,b} \mid \{a, b\} \in \mathcal{K}_{\text{cooccur}}, \ a \in X, \ b \in X\} \quad (3)$$

where each unordered pair $\{a, b\}$ appears only once after deduplication. If no reminders are generated for the input, we set $\mathcal{R}_{\text{AIR}} = \{r_{\text{gen}}\}$, where $r_{\text{gen}}$ is a generic reminder[1].

**Counter-Inertial Reasoning.** All generated reminders in $\mathcal{R}_{\text{AIR}}$ are then injected into the original input $X$ to form an augmented prompt, which is submitted to the LLM $\mathcal{M}$ for faithful inference:

$$y = \mathcal{M}(f_{\text{inject}}(X, \mathcal{R}_{\text{AIR}})) \quad (4)$$

---

[1]Generic template: "Please read the original text carefully, especially paying attention to information that is often assumed by default but not actually given in the question."

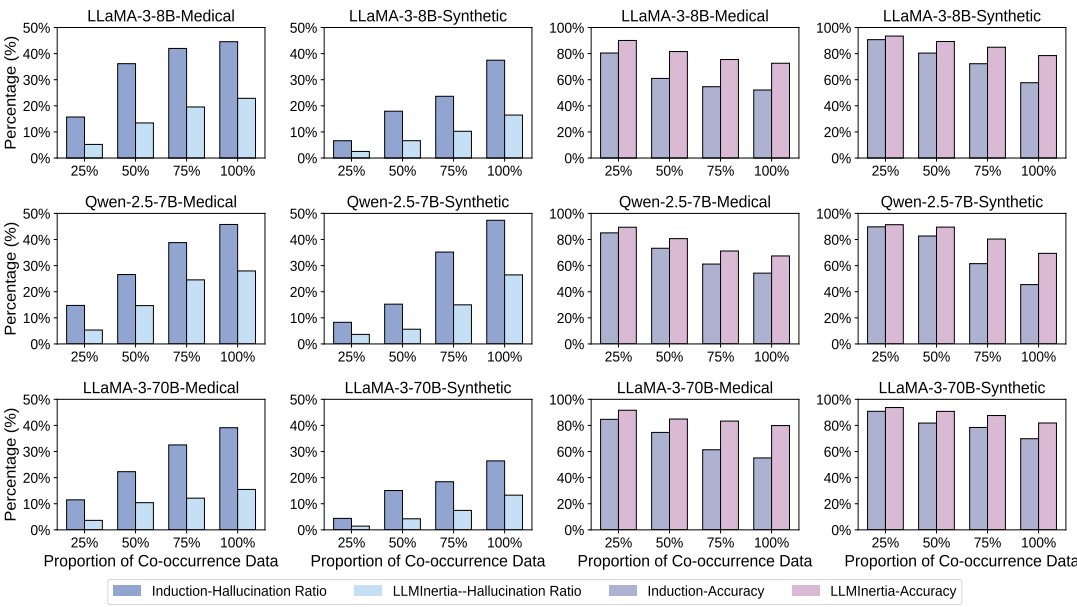

*Figure 4.* Percentage performance (%) of different LLMs on medical and synthetic benchmarks across varying proportions of co-occurrence data. Performance is measured by hallucination induction rate and accuracy after induction, both with and without LLMInertia mitigation.

Here, the injection function $f_{\text{inject}}$ concatenates the adaptive counter-inertial reminders with the original input and is implemented via tailored prompts to the LLM (see Appendix C.1.3). This explicit augmentation steers the model's attention towards potentially misleading inertial associations, promoting more faithful and context-based reasoning while reducing reliance on spurious correlations.

## 4.2. Effectiveness of LLMInertia on Induced Data.

To evaluate the effectiveness of LLMInertia in mitigating input-evidence-unfaithful hallucinations under our constructed induced co-occurrence data settings, we compare the performance of various LLMs after induction, with and without the application of LLMInertia, across two induction paradigms. In the "excessive co-occurrence data increases faithfulness hallucinations" paradigm (Figure 4), LLMInertia consistently and substantially reduces hallucination rates—by up to 23.64%—across all models and both medical and synthetic corpora, while also achieving significant accuracy gains of up to 24.79%. We further examine the "extensive pretraining amplifies faithfulness hallucinations" setting—where the co-occurrence ratio is fixed at 25% and the number of training epochs increases—and find that LLMInertia robustly mitigates hallucinations across epochs, reducing hallucination rates by up to 35% and boosting accuracy by up to 35.68% (see Figure 7 in Appendix B.2 due to space constraints). This confirms the effectiveness of LLMInertia even when cognitive inertia is reinforced through prolonged exposure.

## 4.3. Effectiveness of LLMInertia on Diverse Tasks

To further validate the applicability and effectiveness of LLMInertia in broader scenarios, we conduct large-scale experiments on four context-rich NLP benchmarks that serve as suitable testbeds for evaluating the faithfulness of LLMs to input evidence. We continue to use the three backbone models from previous experiments: LLaMA-3-8B (Dubey et al., 2024), Qwen2.5-7B (Yang et al., 2024a), and LLaMA-3-70B (Dubey et al., 2024). Specifically, we select two summarization datasets (CNN/DailyMail (Chen et al., 2016) and the challenging dialogue-based SAMSum (Gliwa et al., 2019)) and two question answering datasets (SQuAD V2 (Rajpurkar et al., 2016; 2018) and HaluEval (Li et al., 2023), a recently introduced hallucination benchmark). As HaluEval is not publicly available, we reconstruct it according to the original methodology.

**Baselines.** LLMInertia is compared against several representative baselines spanning diverse methods: (1) Base Model: original instruction-tuned backbone; (2) Naive Prompting: using direct instruction-based prompting (see Appendix C.2); (3) Chain-of-thought prompting (CoT): generates intermediate reasoning steps to improve answer quality (Wei et al., 2022); (4) SymbCoT: a chain-of-thought reformulation framing reasoning as symbolic inference to enhance faithfulness (Xu et al., 2024a); (5) Lookback: a hallucination detection and mitigation system leveraging linear classifiers over lookback ratio features (Chuang et al., 2024); (6) Supervised Fine-Tuning (SFT): models further trained on task-specific samples; (7) SelfCheck: the LLM reviews and revises its output for contradictions with the input.

*Table 1.* Evaluation results of faithful hallucinations for all methods on four context-rich summarization and QA datasets, with three LLM backbones of different scales. "Consis" and "AlignS" denote Consistency and AlignScore, respectively.

| Method | Summary Task | | | | QA Task | | | |
| | CNN/Daily Mail | | SAMSum | | SQuAD V2 | | HaluEval | |
| | Consis($\uparrow$) | AlignS($\uparrow$) | Consis($\uparrow$) | AlignS($\uparrow$) | Anah-v2($\downarrow$) | AlignS($\uparrow$) | Anah-v2($\downarrow$) | AlignS($\uparrow$) |
|---|---|---|---|---|---|---|---|---|
| LLaMA-3-Instruct-8B | | | | | | | | |
| Base | 86.40 | 83.28 | 90.79 | 90.67 | 14.32 | 95.94 | 12.48 | 96.35 |
| Prompt | 87.73 | 86.51 | 91.08 | 90.98 | 15.02 | 95.50 | 8.92 | 96.17 |
| CoT | 86.47 | 84.26 | 90.14 | 88.41 | 13.54 | 95.46 | 11.30 | 96.47 |
| SymbCoT | 86.52 | 84.33 | 90.23 | 88.37 | 13.70 | 95.57 | 11.66 | 96.58 |
| Lookback | 86.77 | 84.05 | 89.36 | 88.40 | 13.37 | 95.44 | 11.43 | **96.77** |
| SFT | 90.11 | 84.20 | 89.56 | 87.14 | 13.34 | 95.24 | 8.27 | 96.44 |
| SelfCheck | 87.95 | 86.87 | 90.84 | 90.43 | 14.89 | 95.52 | 10.26 | 96.23 |
| **LLMInertia** | **90.33** | **88.24** | **91.54** | **92.33** | **12.53** | **96.34** | **8.21** | 96.35 |
| LLaMA-3-Instruct-70B | | | | | | | | |
| Base | 89.05 | 85.41 | 92.37 | 91.43 | 11.52 | 96.83 | 10.32 | 96.77 |
| Prompt | 89.36 | 85.63 | 92.54 | 91.33 | 10.26 | 96.24 | 9.55 | 97.01 |
| CoT | 89.02 | 85.49 | 90.97 | 91.43 | 11.15 | 96.13 | 10.06 | 96.65 |
| SymbCoT | 89.06 | 85.57 | 91.04 | 91.52 | 11.38 | 96.47 | 10.23 | 96.76 |
| Lookback | 88.61 | 85.70 | 90.24 | 91.07 | 11.01 | 94.82 | 11.29 | 96.30 |
| SFT | **91.65** | 86.67 | 92.98 | 90.76 | 12.01 | 96.73 | **7.52** | 96.69 |
| SelfCheck | 89.57 | 85.98 | 92.03 | 91.22 | 10.01 | 96.06 | 10.29 | 96.66 |
| **LLMInertia** | 91.40 | **86.75** | **93.02** | **91.97** | **9.56** | **97.04** | 8.33 | **97.22** |
| Qwen-2.5-Instruct-7B | | | | | | | | |
| Base | 83.76 | 82.56 | 85.23 | 86.42 | 17.22 | 93.65 | 14.75 | 95.05 |
| Prompt | 84.51 | 84.83 | 86.06 | 85.41 | 16.31 | 93.93 | 13.98 | 95.14 |
| CoT | 84.34 | 81.97 | 84.22 | 86.71 | 13.06 | 93.58 | 11.23 | 95.33 |
| SymbCoT | 84.45 | 81.84 | 84.31 | 86.72 | 13.25 | 93.67 | 11.38 | 95.35 |
| Lookback | 84.76 | 81.52 | 85.27 | 83.30 | 13.68 | 93.65 | 11.34 | 95.32 |
| SFT | 89.12 | 81.13 | 87.69 | 88.95 | 12.41 | 94.17 | 11.14 | **96.39** |
| SelfCheck | 84.32 | 84.90 | 85.03 | 86.39 | 15.89 | 94.26 | 13.78 | 95.17 |
| **LLMInertia** | **89.31** | **85.40** | **88.03** | **89.14** | **11.22** | **95.56** | **10.47** | 96.10 |

**Metrics.** We employ several targeted metrics for measuring input-evidence-unfaithful hallucination. **AlignScore** (Zha et al., 2023) utilizes a trained alignment model across diverse NLP tasks to reliably detect contradictions or factual inconsistencies in relation to context, supporting both summarization and QA. **UniEval Consistency** (Zhong et al., 2022) is a widely used metric for factual alignment in summarization. **Anah-V2** (Gu et al., 2024) leverages EM-based iterative self-training for hallucination detection by reference-based validation, supporting the QA task.

**Results Analysis.** As shown in Table 1, where only Lookback and SFT require training, other methods are training-free. Across all LLM backbones and datasets, LLMInertia consistently achieves superior performance in minimizing input-evidence-unfaithful hallucinations, outperforming all training-free baselines. For instance, with the Qwen-2.5 backbone, LLMInertia achieves average improvements of 4.27% (Consistency) and 2.72% (AlignScore) on CNN/Daily Mail, 2.63% (Consistency) and 2.87% (AlignScore) on SAMSum, 3.33% (Anah-v2) and 1.72% (AlignScore) on SQuAD v2, and 2.04% (Anah-v2) and 0.71% (AlignScore) on HaluEval. Especially compared to the prompt-based approach, LLMInertia achieves superior performance, as its adaptive counter-inertia reminders explicitly target key and error-prone parts of the input evidence,

delivering more focused guidance than generic prompts.

Furthermore, LLMInertia surpasses strong training-required methods in most cases, including the supervised baseline SFT. However, on the HaluEval dataset, regular and predictable structural patterns lower the evaluation difficulty, thereby narrowing the performance gap between different approaches; as a result, SFT can sometimes achieve results comparable to or slightly better than LLMInertia. In contrast, on more diverse and challenging datasets such as SAMSum—which is conversational, multi-speaker—and SQuAD v2.0—which features a broader range of question types, including unanswerable questions—LLMInertia's adaptive, training-free approach leads to more robust and consistent improvements. Overall, these results demonstrate the effectiveness and generalization capability of LLMInertia across different model architectures and tasks. The error analysis can be found in Appendix B.5.

### 4.4. LLM-based Evaluation

To further verify the reliability of the LLMInertia, we conducted an LLM-based evaluation using GPT-5 to assess the faithfulness of generated results (on a 1–5 scale), and reported the average scores across instances. Table 2 presents evaluation results based on the LLaMA-3-Instruct-8B back-

| An example from a co-occurrence induction dataset. | An example from the SQuAD-v2 summarization dataset. |
|---|---|
| **Context:** The current chief complaint is stomach pain, and no clinical features related to gastric hemorrhage being present, ... 
 **Question:** Does Gastric Hemorrhage exist or hold? 
 **Reminder: stomach pain and gastric hemorrhage may not necessarily be related. Please carefully examine the original text and make a thorough judgment.** 
 **without LLMInteria:** .., Based on typical clinical cases, stomach pain often leads to concerns about more severe issues such as bleeding in the stomach. [Conclusion] Yes. ✗ 
 **with LLMInertia:** ... The reminder highlights that not all cases of stomach pain are caused by bleeding in the stomach...Since there is no direct evidence of gastric hemorrhage, I conclude that gastric hemorrhage does not exist for this patient. 
 [Conclusion] No. ✓ | **Context:** Tumor antigens are presented on MHC class I molecules in a similar way to viral antigens. This allows killer T cells to recognize... 
 **Question:** Tumor antigens are complexed with MHC class I molecules in the same way as what antigens? 
 **Reminder: Tumor antigens and killer T cells may not necessarily be related. Please carefully examine the original text and make a thorough judgment.** 
 **without LLMInteria:** ...Since the context focuses on tumor antigens and their interaction with killer T cells, and MHC class I molecules are presenting tumor antigens...[Conclusion] Tumor antigens. ✗ 
 **LLMInertia:** ...The passage states, "Tumor antigens are presented on MHC class I molecules in a similar way to viral antigens.", This means the method of presenting tumor antigens is being directly compared to how viral antigens are presented. [Conclusion] Viral antigens. ✓ |

*Figure 5.* Real case studies from the Medical induction dataset and the SQuAD-v2 summarization dataset. Each example highlights the key reasoning steps and shows how adaptive counter-inertial reminders guide the LLM to prioritize input evidence over memorized co-occurrences, enabling more faithful reasoning.

*Table 2.* GPT-based evaluation results across four datasets. The last row reports the Spearman correlation with AlignScore.

| Method | Summary Task | | QA Task | |
|---|---|---|---|---|
| | CNN/Daily Mail | SAMSum | SQuAD v2 | HaluEval |
| Base | 4.11 | 4.18 | 4.60 | 4.62 |
| Prompt | 4.26 | 4.25 | 4.63 | 4.57 |
| CoT | 4.18 | 4.24 | 4.62 | 4.69 |
| SymbCoT | 4.21 | 4.21 | 4.58 | 4.64 |
| Lookback | 4.20 | 4.17 | 4.51 | **4.70** |
| SFT | 4.32 | 4.20 | 4.51 | 4.68 |
| SelfCheck | 4.32 | 4.25 | 4.67 | 4.65 |
| **LLMInertia** | **4.54** | **4.58** | **4.79** | 4.68 |
| *Spearman* | *0.766* | *0.623* | *0.635* | *0.669* |

*Table 3.* Ablation results for LLMInertia (%). "Consis" and "AlignS" denote Consistency and AlignScore, respectively.

| Method | CNN/Daily Mail | | SQuAD v2 | |
|---|---|---|---|---|
| | Consis(↑) | AlignS(↑) | Anah-v2 | AlignS(↑) |
| LLMInertia | 90.33 | 88.24 | 12.53 | 96.34 |
| w/o Entity Extraction | 87.54 | 86.58 | 14.98 | 95.63 |
| w/o Inertia Probe | 85.76 | 82.85 | 17.48 | 92.15 |
| w/o Adaptive Reminder | 87.03 | 86.44 | 14.99 | 95.24 |

bone (Complete results for all backbones are shown in Appendix B.4, Table 9.). As observed, LLMInertia achieves the highest average scores across most datasets, demonstrating superior performance in mitigating unfaithful hallucinations and highlighting the robustness of the method. Furthermore, there is a significant Spearman correlation between GPT-based evaluation scores and AlignScore metrics, indicating the reliability and consistency of our evaluation approach.

### 4.5. Ablation Study

We conducted a systematic ablation experiments with LLaMA-3-Instruct-8B. For each ablation, one key component was removed with other settings kept as unchanged as possible. **w/o Entity Extraction**: Entities are not extracted; co-occurrence probing and reminder cannot be performed. Thus, a generic reminder is injected into all samples. **w/o Inertia Probe**: Entities are extracted, but high-risk pairs are not identified via LLM probing. Instead, a sliding window (size=5) determines co-occurrence pairs (Exhaustive pairing was overly noisy; sliding window balances coverage and performance). **w/o Adaptive Reminder**: Upon detecting high-risk pairs, only a fixed generic reminder is injected, not inertia-specific adaptation.

As shown in Table 3, the largest drop occurs without co-

occurrence probing, underscoring its key role in identifying the LLM's internal co-occurrence biases. Removing adaptive reminders leads to a moderate decline, indicating their necessity for focusing the model on error-prone evidence. Omitting entity extraction, which leads to general reminders, results in a slight reduction. Overall, these results demonstrate that all components are essential and complementary for improving evidence faithfulness.

### 4.6. Analysis of Cognitive Inertia-Driven Hallucinations

*Table 4.* Proportion of cognitive inertia-driven hallucinations corrected by each method on CNN/DailyMail and SAMSum.

| Method | CNN/DailyMail (%) | SAMSum (%) |
|---|---|---|
| Prompt | 22.86 | 22.58 |
| SFT | 27.14 | 25.81 |
| CoT | 21.47 | 20.51 |
| SymbCoT | 22.86 | 20.97 |
| Lookback | 20.00 | 19.35 |
| **LLMInertia** | **34.29** | **32.26** |

To quantify the contribution of cognitive inertia among various causes of input-evidence-unfaithful hallucinations in LLMs, we conducted fine-grained manual annotation of 100 low-AlignScore outputs from LLaMA-3-Instruct-8B on both the CNN/DailyMail and SAMSum datasets. Cognitive inertia-driven hallucinations account for 31.08% and 33.79% of all errors, respectively, highlighting its significance as a source of unfaithfulness. For each baseline, we further conducted manual evaluation of effectiveness specif-

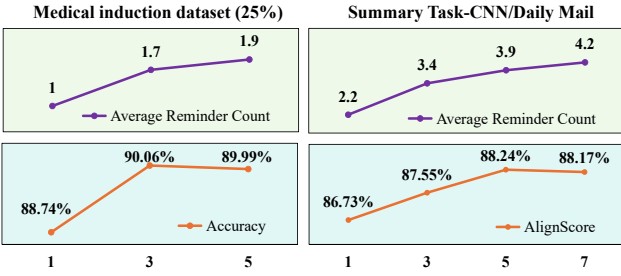

*Figure 6.* Results for the hyperparameter analysis are shown for the Medical dataset (25% induction rate) and the CNN/Daily Mail in the left and right columns, with varying numbers of co-occurring entities returned per entity by the probe function. For each dataset, the top plot shows the average number of counter-inertial reminders, and the bottom shows the performance metric.

ically on these cases, defining the mitigation ratio as the proportion of errors corrected by each method. As shown in Table 4, LLMInertia achieves the highest mitigation ratio on both datasets, substantially outperforming other approaches and confirming its targeted effectiveness.

### 4.7. Case Study

We include a case study to further illustrate the effectiveness of LLMInertia. Figure 5 presents representative examples from both the Medical dataset (25% induction rate) and the SQuAD-v2 summarization dataset. Each example highlights the key reasoning steps and demonstrates how adaptive counter-inertial reminders help the LLM shift its focus from internal co-occurrence biases to input evidence. This verifies that LLMInertia enables the LLM to overcome cognitive inertia, perform counter-inertial reasoning, and make more faithful judgments aligned with the input context.

### 4.8. Hyperparameter Analysis of the Probe Function

We perform a hyperparameter analysis to assess how the number of co-occurring entities returned per entity by the probe function affects both the number of counter-inertial reminders generated and model performance. As shown in Figure 6, for the Medical induction dataset (left column), increasing this number raises the average count of counter-inertial reminders (top plot), but accuracy peaks at 3 and declines slightly at 5 (bottom plot), indicating that 3 provides the best trade-off between coverage of input-related cognitive inertia and avoidance of excessive noise. For the CNN/Daily Mail summarization dataset (right column), setting this value to 5 yields the highest AlignScore. Therefore, we set the hyperparameter to 3 for induction corpora and to 5 for summarization and QA tasks in our experiments.

### 4.9. Scalability of LLMInertia

To validate the scalability of LLMInertia in scenarios with longer input texts, we benchmarked its latency (in seconds) on LLaMA-3-Instruct-8B, empirically comparing it to the

*Table 5.* Average latency overhead (seconds). The last two columns show the times increase over the baselines.

| Input | Base | Prompt | LLMInertia | LLMInertia vs. Base | LLMInertia vs. Prompt |
|---|---|---|---|---|---|
| 500 | 0.5 | 0.6 | 1.2 | 2.2x | 2.1x |
| 1000 | 1.2 | 1.3 | 3.1 | 2.58x | 2.38x |
| 2000 | 1.7 | 1.8 | 3.7 | 2.18x | 2.05x |
| 4000 | 2.6 | 2.9 | 5.6 | 2.15x | 1.93x |

base and prompt methods. The evaluation was conducted using input texts of varying lengths (500-2000 tokens from CNN/Daily Mail, and 4000 tokens from LongBench(Bai et al., 2024)). The results are presented in Table 5.

We observe that LLMInertia's latency overhead remains moderate as input length increases, with the relative gap to baselines first rising and then falling. For inputs of 500–1000 tokens, the low baseline latency makes pipeline scheduling comparatively costly, resulting in an increased overhead ratio. As input length grows (1000 → 2000 → 4000), the baseline latency increases accordingly, while LLMInertia's cost increases more slowly—mainly depending on the number of extracted entities (e.g., the extraction step lists all important entities, and the probing step returns the most closely related entities; see Appendix C.1). Thus, the overhead is increasingly amortized over longer contexts, with the ratio dropping. These results demonstrate that LLMInertia's overhead growth is manageable and supports efficient scalability for long-input scenarios.

## 5. Conclusion and Limitations

In this paper, we conduct the first systematic investigation into how excessive co-occurrence associations within parametric priors trigger input-evidence-unfaithful hallucinations, and quantify their influence on a model's resistance to input evidence. Building on these findings, we identify cognitive inertia as a central cause of such hallucinations in LLMs and introduce LLMInertia, an adaptive counter-inertial reasoning framework supporting targeted mitigation. Extensive experiments on co-occurrence induction datasets, as well as four summarization and QA task datasets, demonstrate the superiority and robustness of LLMInertia, advancing the development of more reliable and faithful LLMs.

This work has two main limitations. First, although input-evidence-unfaithful hallucinations are prevalent in both general and domain-specific applications, our evaluation is limited to general-domain settings due to the lack of publicly available domain-specific datasets for this phenomenon. Second, while our approach is effective for common explicit co-occurrence associations, handling rarer and more complex cases, such as implicit evidence, remains an open direction. Future work will explore graph-based or advanced bias detection methods to further enhance LLMInertia.

## Acknowledgements

This work was supported by the Noncommunicable Chronic Diseases-National Science and Technology Major Project (Grant No. 2023ZD0506501), the Fundamental Research Funds for the Central Universities in UIBE(25QD16), the Guangxi Key Research and Development Program (GuiKe FN2600640478), and the Key Research and Development Program of NingXia (2025BEG02001). We would like to thank the clinicians who participated in the validation and the anonymous reviewers for their valuable feedback.

## Impact Statement

This work studies input-evidence-unfaithful hallucinations in LLMs. We identify *cognitive inertia*: an over-reliance on pretrained co-occurrence associations that causes models to resist conflicting input evidence. Based on this insight, we propose LLMInertia, a training-free, inference-time prompting method that injects adaptive reminders to promote evidence-grounded generation. By reducing such hallucinations, our approach can improve the reliability of LLMs in context-rich and high-stakes applications.

The datasets and prompts used are built based on public medical datasets and are publicly available to facilitate further research. We have not identified any potential negative ethical consequences requiring further consideration.

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

# A. Datasets

## A.1. Entity Pairs

Two categories of entity pairs were constructed, each comprising 40 pairs: (1) realistic medical pairs, identified via statistical analysis of high-frequency co-occurrences in the MIMIC dataset (Johnson et al., 2016) and expert screening to exclude direct causal relationships; and (2) synthetic pairs, automatically generated by Deepseek-R1 (Guo et al., 2025), with entirely fictional entity names that preclude any semantic or causal relationship. The complete list of entity pairs is presented in Table 6.

*Table 6.* Medical entity pairs and fictional entity pairs

| Medical entity pair | Fictional entity pair |
|---|---|
| (Headache, Hypertension) | (Blue Map, Rubber Escalator) |
| (Cough, Pulmonary Tuberculosis) | (Wooden Sky, Mirror Flower) |
| (Fatigue, Hepatitis) | (Rotating Bell, Red Ocean) |
| (Vomiting, Gastric Cancer) | (Invisible Staircase, Scented Shadow) |
| (Chest tightness, Myocardial Infarction) | (Stone Smile, Paper Rain) |
| (Low back pain, Renal Calculus) | (Gummy Desert, Star Key) |
| (Fever, Leukemia) | (Quiet Engine, Sand Time) |
| (Sweating, Hyperthyroidism) | (Mirror Road, Grey Leaf) |
| (Dizziness, Cerebral Infarction) | (Magnetic Flame, Virtual Cup) |
| (Abdominal distension, Gastric Cancer) | (Folding Cat, Electric Feather) |
| (Hematuria, Urinary Tract Infection) | (Blue Gear, Air Station) |
| (Blurred vision, Diabetes Mellitus) | (Reflective Banana, Cotton Bone) |
| (Rash, Systemic Lupus Erythematosus) | (Glass Kite, Cement Ribbon) |
| (Tremor, Parkinson Disease) | (Sound Bubble, Motionless Umbrella) |
| (Palpitation, Arrhythmia) | (Salty Ray, Transparent Note) |
| (Tinnitus, Hypertension) | (Capsule Door, Flowing Bulb) |
| (Sore throat, Rhinitis) | (Dry Subway, Lighthouse Tea) |
| (Myalgia, Influenza) | (Ice Train, Password Cloud) |
| (Somnolence, Liver Dysfunction) | (Jetlag Fish, Silk Manhole Cover) |
| (Loss of appetite, Gastric Ulcer) | (Plastic Ear, Soft Overpass) |
| (Abdominal pain, Cholecystitis) | (Dust Guitar, Tile Cake) |
| (Cough, Heart Failure) | (Silent Clock, Text Rope) |
| (Constipation, Hypothyroidism) | (Flipped Shoe, Feather Glove) |
| (Dyspnea, Lung Cancer) | (Orange Library, High-heel Abacus) |
| (Weight loss, Benign Tumor) | (Sticky TV, Green Towel) |
| (Leg swelling, Deep Vein Thrombosis) | (Scented Paper, Wind Speed Book) |
| (Diarrhea, Colorectal Cancer) | (Round Phone, Warm Coin) |
| (Stomach pain, Gastric Hemorrhage) | (Origami Bed, Magnet Mirror) |
| (Panic, Hypoglycemia) | (Candy Map, Paper Lamp) |
| (Back pain, Pancreatitis) | (Milk Clock, Grey Ringtone) |
| (Dry throat, Diabetes Mellitus) | (Embroidered Remote, Plastic Helmet) |
| (Diminished sense of taste, COVID-19 Infection) | (Note Window, Tasteless Salad) |
| (Menstrual irregularity, Polycystic Ovary Syndrome) | (Soft Light Stamp, Jumping Bottle) |
| (Edema, Nephrotic Syndrome) | (Fog Password, Turning Key) |
| (Pruritus, Liver Cirrhosis) | (Corner Moon, Loud Glass) |
| (Sleep disorder, Anxiety Disorder) | (Floating Sofa, Lazy Seed) |
| (Dysphagia, Esophageal Cancer) | (Sweet Tire, Sleeping Candle) |
| (Numbness of limbs, Cervical Spondylosis) | (Blueberry Sensor, Sonic Brush) |
| (Depressed mood, Alzheimer's Disease) | (Gravity Clock, Sponge Phone) |
| (Hoarse voice, Laryngeal Cancer) | (Foam Nail, Shadow Cutlery) |

## A.2. Co-occurrence Induction Templates

To construct the co-occurrence induction pre-training corpus, 100 diverse templates were generated using Deepseek-R1, capturing a wide range of association scenarios between entities A and B in medical contexts. Entities A and B are replaced with either medical or fictional pairs from Table 6. Due to space constraints, we present a selection of sample templates in Table 7. The full list of templates will be published alongside the code repository.

*Table 7.* Co-occurrence Induction Corpora Construction Templates

| | |
|---|---|
| 1 | "In the diagnostic process, A and B are recorded and analyzed together. |
| 2 | The medical team pays close attention to both symptom A and indicator B. |
| 3 | The hospital database identified A and B as high-frequency co-occurring pairs in this disease and provided doctors with relevant risk alerts. |
| 4 | The simultaneous occurrence of A and B has been reported in a large number of case reports. |
| 5 | Medical institutions have incorporated A and B into the comorbidity screening process. |
| 6 | A and B respectively represent different dimensions of patient's clinical manifestations and laboratory tests. |
| 7 | In clinical pathways, physicians categorize A and B under similar process items. |
| 8 | This is not the first co-occurrence of A and B for the patient; similar patterns have been documented in previous case records, suggesting an intrinsic pathological feature for the individual. |
| 9 | In the study, both types of manifestations, A and B, were used as factors for multivariate analysis, aiming for a comprehensive assessment of patient conditions. |
| 10 | Multiple clinicians, through retrospective analysis of recent hospital cases, found that the co-occurrence frequency of A and B is significantly higher than random background, suggesting a possible pathological mechanism. |
| 11 | Through electronic medical record review, it was found that from the initial visit to admission, A and B persisted together with no apparent temporal delay. |
| 12 | Research papers have previously indicated that the A-B combination is a high-incidence pattern in this disease spectrum, and this case confirms the clinical consistency of that observation. |
| 13 | During hospitalization, both A and B were observed and recorded simultaneously. |
| 14 | The patient's medical history entry system automatically linked the simultaneous existence of A and B and recommended referencing management procedures from previous similar cases. |
| 15 | Physicians noted a stable co-occurrence of symptom A and test result B during diagnosis. |
| 16 | Although A and B represent the patient's subjective experience and objective test data respectively, in actual clinical practice these two often co-occur and affect disease assessment and treatment strategies. |
| 17 | Retrospective patient data show a stable correlation between A and B. |
| 18 | Case studies indicate a high correlation between A and B. |
| 19 | In the multidisciplinary consultation report, the medical team explicitly stated that A and B are two important dimensions for assessing disease severity and must be considered jointly. |
| 20 | During preliminary outpatient screening, the patient reported discomfort of A, and both serum and imaging tests supported abnormal B, suggesting a linked pathological feature. |
| 21 | Medical reports indicate that both A and B have significant diagnostic value as clinical manifestations. |
| 22 | During the patient's treatment, A led to a follow-up visit, and at that visit, B abnormality was coincidentally detected, suggesting that the former may have a warning role. |
| 23 | Clinical experience suggests that the co-occurrence of A and B is a typical diagnostic combination. |
| 24 | During several days of inpatient observation, manifestation A remained significant, and relevant test results also showed abnormal B indices. The two were classified by physicians as common co-occurring factors for this disease. |
| 25 | Physicians found that A and B often occur together during clinical consultation. |

## A.3. Test Set Construction Templates

To construct the test dataset, 25 semantically diverse templates were designed around the theme "A is present, A and B are commonly associated, but B is absent." followed by the query "Is B present?" by Deepseek-R1 requiring a binary (yes/no) model response. The complete set of test templates is provided in Table 8.

*Table 8.* Test Datasets Construction Templates

| No. | Template |
|---|---|
| 1 | The patient's main complaint during this visit is A, which can sometimes indicate B, although no clinical or laboratory abnormalities related to B have been found. |
| 2 | Symptom A is apparent and, while examinations revealed no indications related to B, A is often associated with B in clinical settings. |
| 3 | The patient currently presents with persistent A, and although no changes related to B have been observed, such symptoms are occasionally linked to B. |
| 4 | A occurred, and even though the relevant parameters and examinations revealed no evidence of B, A is considered by some to be a typical presentation of B. |
| 5 | The main issue for this admission is A, and although no manifestations of B were found during the investigation, A may, in certain cases, imply underlying B. |
| 6 | The current chief complaint is A, and despite no clinical features related to B being present, A is sometimes a precursor to B-related conditions. |
| 7 | Clinical symptoms are mainly A, and although evaluation revealed no evidence of B, it is possible that A is an early sign of B. |
| 8 | The patient reported discomfort of A, and examination ruled out B and other related diseases; nevertheless, A may occasionally precede B. |
| 9 | A is relatively prominent, and there is currently no evidence suggesting the presence of B, but A is frequently described in the context of B. |
| 10 | Symptom A appeared during this visit, and while related examinations showed no co-occurrence with B, A could potentially suggest B in some scenarios. |
| 11 | The patient's most prominent symptom is currently A, with no positive signs or test results related to B, but A sometimes occurs alongside B. |
| 12 | A preliminary diagnosis has been made for symptom A, and no accompanying B was found, though A has been known to be an initial sign of B in some instances. |
| 13 | The patient developed A, and even if no detection of B occurred during evaluation, certain cases have reported A in conjunction with B. |
| 14 | A manifestation appeared, and although the examination data do not support the presence of B, A has occasionally been linked with B in literature. |
| 15 | Clinical evaluation revealed prominent symptom A, and while relevant tests excluded the possibility of B, A might still indicate a risk for B. |
| 16 | In this case, A has occurred, and although there is no evidence of B lesion or dysfunction, A could theoretically be related to undetected B. |
| 17 | A is the chief complaint, and the assessment result for B is negative; however, A is often monitored as a potential indicator of B. |
| 18 | A was clearly recorded during this visit, and though no physical signs or abnormal tests related to B were found, the presence of A is occasionally suspicious for B. |
| 19 | The main symptom is A, and B is not present, but A is recognized in some studies as possibly associated with B. |
| 20 | The patient visited due to A, and although examinations excluded B, A should not be entirely dismissed as unrelated to B. |
| 21 | The chief complaint is A, and while no pathological changes related to B were found in various tests, similar symptoms have occurred with B. |
| 22 | The current manifestation is A, with no evidence of B found in relevant system examinations, but patients with A sometimes develop B later. |
| 23 | A persists, and although the result for B is negative, there have been cases where persistent A later revealed B. |
| 24 | The main discomfort during this visit is A, and even though no B was observed during the consultation, persistent A may warrant consideration of B. |
| 25 | Symptom A occurred without B, and laboratory and imaging results were normal, yet it is not uncommon for B to be missed initially when A presents. |

# B. Supplement to the Experimental Section

## B.1. Implementation Details

**Hyperparameters.** We conduct continued pre-training for LLaMA-3-8B (Dubey et al., 2024) and Qwen-2.5-7B (Yang et al., 2024a) using full-parameter tuning, and for Meta-LLaMA-3-70B (Dubey et al., 2024) using parameter-efficient training with LoRA (Hu et al.). We used AdamW optimizer with a batch size of 16, cosine learning rate schedule (initial LR 1e-5, 10% warmup). For LoRA, rank is set to 64, scaling factor $\alpha$ to 16, with adapters applied to all trainable linear layers.

**Software.** All model training and inference are performed using LLaMA-Factory[2].

**Inference.** We employ the vLLM (Kwon et al., 2023) engine for batched model inference under greedy decoding. For each prompt, outputs are generated five times with different seeds and metrics are reported as the average over these generations.

**Computation overhead.** Experiments on LLaMA-3-8B and Qwen-2.5-7B are conducted with a single NVIDIA A100 GPU (80GB), while experiments on Meta-LLaMA-3-70B use four NVIDIA A100 GPUs (80GB each). The total computational overhead is approximately 232 GPU hours, calculated as the sum of wall-clock hours multiplied by the number of GPUs for each experiment.

**Code.** Our code is available at: `https://github.com/THUMLP/LLMInertia`.

## B.2. Supplementary Results: Hallucination Induction and Mitigation Across Training Epochs

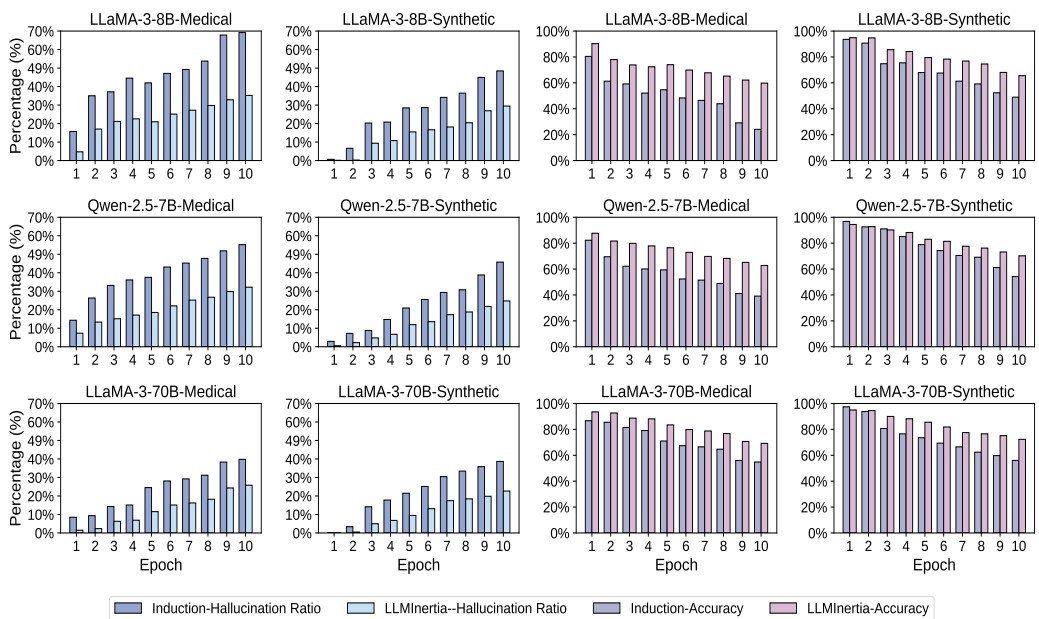

*Figure 7.* Percentage performance (%) of different LLMs on medical and synthetic benchmarks, with the proportion of co-occurrence data fixed at 25%. Performance is measured by hallucination induction rate and accuracy after induction, both with and without LLMInertia mitigation. Results are reported as bar charts for each model and data type, across 10 training epochs.

## B.3. Details of Entity Extraction Evaluation

To assess the reliability of entity extraction process, we conducted evaluation on outputs from the LLaMA3-Instruct-8B model using the CNN/DailyMail and SQuAD v2 datasets. For each dataset, we randomly sampled 50 cases and compared the extracted entities with two sets of references: DeepSeek R1 results and human annotations. Entity extraction performance was measured using F1 score, where an entity was considered correct if successfully identified, regardless of its type. This lenient criterion led to high F1 scores—CNN/DailyMail: 90.49% (DeepSeek R1) vs. 87.62% (human); SQuAD v2: 89.54% (DeepSeek R1) vs. 87.11% (human). This evaluation demonstrates that relying solely on the LLM's capabilities enables

---

[2]`https://github.com/hiyouga/LLaMA-Factory`

effective entity extraction, providing a solid foundation for subsequent steps.

## B.4. Supplementary Results: GPT-based evaluation results

*Table 9.* GPT-based evaluation results across four datasets, using three different LLM backbones. The last row reports the Spearman correlation with AlignScore

| Method | Summary Task | | QA Task | |
|---|---|---|---|---|
| | **CNN/Daily Mail** | **SAMSum** | **SQuAD v2** | **HaluEval** |
| **LLaMA-3-Instruct-8B** | | | | |
| Base | 4.11 | 4.18 | 4.60 | 4.62 |
| Prompt | 4.26 | 4.25 | 4.63 | 4.57 |
| CoT | 4.18 | 4.24 | 4.62 | 4.69 |
| SymbCoT | 4.21 | 4.21 | 4.58 | 4.64 |
| Lookback | 4.20 | 4.17 | 4.51 | **4.70** |
| SFT | 4.32 | 4.20 | 4.51 | 4.68 |
| SelfCheck | 4.32 | 4.25 | 4.67 | 4.65 |
| **LLMInertia** | **4.54** | **4.58** | **4.79** | 4.68 |
| *Spearman* | *0.766* | *0.623* | *0.635* | *0.669* |
| **LLaMA-3-Instruct-70B** | | | | |
| Base | 4.20 | 4.26 | 4.64 | 4.64 |
| Prompt | 4.12 | 4.30 | 4.63 | 4.66 |
| CoT | 4.31 | 4.33 | 4.62 | 4.63 |
| SymbCoT | 4.19 | 4.40 | 4.67 | 4.69 |
| Lookback | 4.28 | 4.28 | 4.59 | 4.60 |
| SFT | 4.39 | 4.24 | 4.68 | 4.64 |
| SelfCheck | 4.41 | 4.27 | 4.67 | 4.66 |
| **LLMInertia** | **4.65** | **4.67** | **4.77** | **4.73** |
| *Spearman* | *0.810* | *0.683* | *0.599* | *0.590* |
| **Qwen-2.5-Instruct-7B** | | | | |
| Base | 4.22 | 4.11 | 4.32 | 4.52 |
| Prompt | 4.33 | 4.10 | 4.37 | 4.54 |
| CoT | 4.13 | 4.16 | 4.27 | 4.60 |
| SymbCoT | 4.11 | 4.18 | 4.33 | 4.59 |
| Lookback | 4.15 | 4.15 | 4.29 | 4.52 |
| SFT | 4.16 | 4.24 | 4.38 | **4.65** |
| SelfCheck | 4.35 | 4.20 | 4.30 | **4.50** |
| **LLMInertia** | **4.42** | **4.46** | **4.52** | 4.63 |
| *Spearman* | *0.786* | *0.762* | *0.743* | *0.826* |

## B.5. Error Analysis

We summarize key observations from our error analysis:

- **Probing Failures:** The probing stage can sometimes fail to accurately capture the model's inertia entities, either by missing the most over-learned associations or by error propagation from incomplete or imprecise entity extraction, resulting in missed key reminders or distracting ones.

- **Stubborn Cognitive Inertia:** For some deeply embedded co-occurrence associations, even accurate adaptive reminders may be ignored, and hallucinations persist.

- **Oscillating in Complex Cases:** In certain complex semantic or reasoning scenarios, the model may initially respond to the reminder but later revert during inference, oscillating and ultimately producing unfaithful answers.

# C. Introduction to the Prompt Used

## C.1. Instruction Description for LLMInertia

### C.1.1. ENTITY EXTRACTION

Given a text input, we prompt the LLM to extract all entities. The instruction is as follows:

> **Instruction:** Carefully read the following input and list all important entities, in the format below.
> - Output entities as a comma-separated list. No extra words.
>
> **Example:**
> Input: The patient reports experiencing a persistent cough and fever for three days. Chest X-ray was performed with no abnormalities detected.
> Output: Cough, Fever, three days, Chest X-ray
>
> **Input:** [Your Input]
> **Output:**

### C.1.2. CO-OCCURRENCE PROBE

For each extracted entity, we prompt the LLM to output the three most closely related entities, in the format of co-occurrence pairs. The instruction is as follows:

> **Instruction:** For each given entity, output three entities that are most closely related to it. List all results in the form of tuple. See the format and example below.
>
> **Format:**
> Entities: Entity1, Entity2
> Output:
> (Entity1, Co1), (Entity1, Co2), (Entity1, Co3)
> (Entity2, Co1), (Entity2, Co2), (Entity2, Co3)
>
> **Example:**
> Entities: Headache, Cough
> Output:
> (Headache, Dizziness), (Headache, Nausea), (Headache, Chest pain)
> (Cough, Shortness of breath), (Cough, Sore throat), (Cough, Fatigue)
>
> **Entities**:[Your Entities]
> **Output:**

### C.1.3. COUNTER-INERTIAL REASONING

Finally, we prompt the LLM to answer the given question, explicitly encouraging evidence-grounded reasoning with the adaptive reminder inserted. The response format is strictly enforced. The instruction is as follows:

> **Instruction:** Please answer the following question and strictly respond in the format: [Conclusion]: Yes/No.
>
> **Here are two examples:**
>
> *Example 1:*
> Question: The patient's main complaint during this visit is Headache, which can sometimes indicate Hypertension,

although no clinical or laboratory abnormalities related to Hypertension have been found. Does Hypertension exist or hold?

Reminder: Headache and Hypertension may not necessarily be related. Please carefully examine the original text and make a thorough judgment.

Answer: [Conclusion] No.

*Example 2:*

Question: The patient has been experiencing persistent cough for several days, which can sometimes suggest Pneumonia; imaging results showed clear lungs with signs of infection Pneumonia. Does Pneumonia exist or hold?

Reminder: Cough and Pneumonia may not necessarily be related. Please carefully examine the original text and make a thorough judgment.

Answer: [Conclusion] Yes.

**Question:** [Your question]
**Reminder:** [Adaptive reminder]
**Answer:**

## C.2. The Prompt for the Base and Prompt Versions Used for Model Testing

*Table 10.* For the summary and QA tasks, show the instructions for the base and prompt versions of the three model bases.

| Tasks | Base | Prompt |
|---|---|---|
| Summary | Write a summary of the following news. | Write a summary of the following news. Attention should be paid to the faithfulness of the abstract with the original text to avoid generating content with hallucinations. |
| QA | You are a question answerer. You should answer the questions directly based on the given reference without adding any prefixes or suffixes, and without analyzing the answers. After answering the question, do not say anything else. Reference document: ...... Please answer the question based on the above reference: | You are a question answerer. You should answer the questions directly based on the given reference without adding any prefixes or suffixes, and without analyzing the answers. After answering the question, do not say anything else. Please do not output content that is inconsistent with the context, and avoid giving irrelevant or contradictory answers. Reference document: ...... Please answer the question based on the above reference: |

