# OpenReview forum: "LLMInertia: Adaptive Counter-Inertial Reasoning to Improve Evidence Faithfulness in Large Language Models"
_ICML.cc/2026/Conference — ICML 2026 regular_

### Official Review · Reviewer_sMJH · 2026-02-16

**Soundness:** 2
**Presentation:** 3
**Significance:** 2
**Originality:** 2
**Overall Recommendation:** 3
**Confidence:** 4

**Summary:**

This work identifies cognitive inertia in large language models, an overreliance on learned cooccurrence patterns that leads to input unfaithful hallucinations. And proposes LLMInertia, an adaptive counter-inertial prompting framework that mitigates such errors.

**Compliance With Llm Reviewing Policy:**

Affirmed.

**Final Justification:**

I appreciate the additional experiments conducted. From the results, SelfCheck appears to be a strong baseline; for example, it achieves 84.9, which is higher than all other baselines reported in the paper. Therefore, it is necessary to compare against this baseline across all datasets rather than only a subset. Similarly, I believe a comprehensive evaluation of LLM-based evaluation methods is also necessary. In addition, the missing related work should be included in the submission version.

**Key Questions For Authors:**

The manual evaluation selects 100 samples with low AlignScore. Are all of these 100 samples actually unfaithful? It would be helpful to clarify whether these examples were verified to contain factual inconsistencies, and how representative they are of the overall error distribution.

**Limitations:**

yes

**Strengths And Weaknesses:**

Strengths

1. This work addresses the important problem of unfaithful generation in language models, which is highly relevant in many real world applications.

2. The paper identifies the issue that models may overly rely on high frequency cooccurrence patterns learned during pretraining, which provides a plausible explanation for certain types of unfaithful outputs.

3. The authors propose a mitigation strategy based on cooccurrence relationships, offering a simple and practical approach to reduce such errors.

Weaknesses

1. The problem of unfaithfulness has been extensively studied in prior language model research. However, the paper mainly cites recent LLM work and omits several highly relevant related works in abstractive summarization and factual consistency, such as: Enhancing Factual Consistency of Abstractive Summarization, CLIFF: Contrastive Learning for Improving Faithfulness and Factuality in Abstractive Summarization, Towards Improving Faithfulness in Abstractive Summarization. The lack of discussion of these closely related works weakens the positioning of the contribution.

2. While cooccurrence bias is indeed one source of unfaithful outputs, there are many other causes of unfaithfulness, such as reasoning errors, overgeneralization, misinterpretation of context, and decoding artifacts. Therefore, from a practical perspective, the overall impact of this work may be limited if it only addresses one specific source of the problem.

3. The paper lacks a strong and direct baseline, such as returning the generated output to the LLM and asking it to explicitly check for contradictions with the input, then filtering or revising inconsistent content. I believe such a simple self checking strategy could potentially resolve many unfaithful cases and would provide a stronger comparison.

4. The evaluation does not use LLM based judging and instead relies on traditional learned metrics. Although these metrics are correlated with faithfulness, they are not perfectly aligned with human judgment. Why not use a state of the art LLM as an evaluator? This could improve both reliability and interpretability of the results.

---

> ### Author Rebuttal · Authors · 2026-03-30
>
> We are truly grateful for your constructive feedback. Below, we address each of the concerns you raised.
>
> **1.Related Work:** We thank the reviewer for highlighting these foundational and influential works on factual faithfulness in abstractive summarization.
> - **Enhancing Factual Consistency:** Models facts as relational graphs and integrates them into a transformer encoder-decoder via graph attention, improving summary factuality.
> - **CLIFF:** Uses contrastive learning with reference and negative samples to better distinguish facts from hallucinations.
> - **Towards Improving Faithfulness:** Identifies unfaithfulness from limited source understanding and fluency bias. Proposes multi-task training and max-margin loss to reduce overconfidence.
>
> While prior work has laid a strong foundation for addressing unfaithfulness in language models, our study delves deeper into the mechanisms behind input-unfaithful hallucinations in LLMs. We are the first to systematically quantify how strong co-occurrence associations in LLM memory lead to resistance to input evidence—termed "cognitive inertia." We further mitigate inertia by adaptively injecting reminders during inference, offering a task-agnostic, training-free solution for context-rich generation tasks.
>
> **2. Practical Impact of LLMInertia:** We fully agree that unfaithful outputs in LLMs have diverse causes. Cognitive inertia stems from the causal language modeling objective, which encourages statistical regularities over deep semantic understanding [1]. As such, it is a fundamental source and may also help illuminate other causes or failures, such as reasoning errors.
> To assess practical significance, we performed error attribution and found that cognitive inertia-driven hallucinations account for about 1/3 of all input-unfaithful errors (Section 4.5, Table 2). Moreover, our method outperforms all baselines in correcting this error type by 6.45%–14.29%. These results highlight the importance and practicality of our approach.
> > [1] Kang, C., et al. Impact of co-occurrence on factual knowledge of large language models. EMNLP 2023.
>
> **3. Self-Check Baseline:** As recommended, we implemented a self-checking baseline where the LLM reviews and revises its output for contradictions with the input.
>
> |Methods|CNN/Daily Mail (Consistency↑)|CNN/Daily Mail (AlignScore↑)|SQuAD v2 (Anah-v2↓)|SQuAD v2(AlignScore↑)|
> |-|-|-|-|-|
> | LLaMA-3-Instruct-8B|
> |Self Check|87.95|86.87|14.89|95.52|
> |LLMInertia|90.33|88.24|12.53|96.34|
> |LLaMA-3-Instruct-70B|
> |Self Check|89.57|85.98|10.01|96.06|
> |LLMInertia|91.4|86.75|9.56|97.04|
> |Qwen-2.5-Instruct-7B|
> |Self Check|84.32|84.9|15.89|94.26|
> |LLMInertia|89.31|85.4|11.22|95.56|
>
> As shown, LLMInertia clearly outperforms the self-check baseline across three backbones and two datasets. This is because cognitive inertia arises from deeply embedded parametric memory that instructions alone cannot easily correct (as supported by [1]). Larger models, with more persistent memorization, also gain little from self-checking [2]. In contrast, LLMInertia proactively identifies cognitive inertia risks and injects adaptive reminders before inference, providing a targeted and effective improvement in faithfulness.
> > [1] Huang J, et al. LLMs Cannot Self-Correct Reasoning Yet. ICLR, 2024.
> > [2] Carlini N, et al. Quantifying memorization across neural language models. ICLR, 2022.
>
> **4. LLM-based Evaluation:** We prioritized widely used automatic metrics for their efficiency, as LLM-based evaluation can be more costly and may contain errors.
> We additionally used GPT-5 to score the faithfulness of summaries (1–5 scale) on 100 random examples per dataset. As shown below, LLMInertia achieved the highest faithfulness scores, and LLM-based and automatic metrics showed strong consistency.
>
> |Method|CNN/Daily Mail|SAMSum|
> |-|-|-|
> |Base|4.14|4.2|
> |Prompt|4.23|4.26|
> |CoT|4.16 |4.21|
> |SymbCoT|4.2|4.23|
> |Lookback|4.17|4.14|
> |SFT|4.35|4.20|
> |LLMInertia|4.56|4.57|
>
> *Spearman correlation with AlignScore: 0.714(CNN/Daily Mail) / 0.613(SAMSum)*
>
> **5. Discussion of Manual Evaluation:** Low-AlignScore samples were treated as “suspected hallucinations” and manually verified, with factual inconsistencies confirmed in 87/100 CNN/Daily Mail and 88/100 SAMSum cases. We then performed manual error classification, counting each hallucinated sentence separately. The results are shown below:
>
> |Error Type|CNN/Daily Mail (%)|SAMSum (%)|
> |-|-|-|
> |Cognitive Inertia|31.08|33.79|
> |Contextual Misinterpretation|25.47|17.49|
> |Reasoning/Integration Error|26.44|33.74|
> |Extrinsic Hallucination/Fabrication|12.80|9.41|
> |Others|4.21|5.57|
>
> While full annotation would yield a complete error distribution, our use of context-rich datasets, manual verification, and a clear classification scheme together ensure the reliability of our analysis, confirming that cognitive inertia is a significant cause of unfaithful hallucinations in LLMs.
>
> We will incorporate these discussions into the revised paper.

---

> > ### Author Rebuttal · Reviewer_sMJH · 2026-04-03
> >
> > I appreciate the additional experiments conducted. From the results, SelfCheck appears to be a strong baseline; for example, it achieves 84.9, which is higher than all other baselines reported in the paper. Therefore, it is necessary to compare against this baseline across all datasets rather than only a subset. Similarly, I believe a comprehensive evaluation of LLM-based evaluation methods is also necessary. In addition, the missing related work should be included in the submission version.

---

> > > ### Author Response · Authors · 2026-04-06
> > >
> > > Thank you very much for the time and effort you have devoted to reviewing our submission. Your continued feedback has been extremely valuable in helping us improve the quality of the manuscript. We sincerely value your concerns and are committed to engaging with these issues thoughtfully.
> > >
> > > **1.Self-Check Baseline.** Thank you for your follow-up suggestion. As requested, we extended the self-check baseline experiments to all datasets and models. The results (see table below) consistently show that while self-checking is a competitive baseline, LLMInertia systematically outperforms it in all settings. Your valuable advice made our experimental comparison more comprehensive and complete, and we will include these significant results in the revised paper.
> > >
> > >
> > > |  Methods    |CNN/DailyMail  (Consistency↑)|CNN/DailyMail  (AlignScore↑)|  SAMSum  (Consistency↑) |  SAMSum  (AlignScore↑)|  SQuAD v2  (Anah-v2↓)|SQuAD v2  (AlignScore↑)|HaluEval  (Anah-v2↓)|HaluEval     (AlignScore↑)|
> > > |--|----|----|----|---|---|---|-----|----|
> > > | **LLaMA-3-Instruct-8B** |
> > > |    SelfCheck     |87.95|86.87|90.84|90.43|14.89|95.52|10.26|96.23|
> > > |    LLMInertia    |90.33|88.24|91.54|92.33|12.53|96.34|8.21|96.35|
> > > |   **LLaMA-3-Instruct-70B**   |  | |     | | | | ||
> > > |    SelfCheck    |89.57|85.98|92.03|91.22|10.01|96.06|10.29|96.66|
> > > |    LLMInertia            |91.40|86.75|93.02|91.97|9.56|97.04|8.33|97.22|
> > > |  **Qwen-2.5-Instruct-7B**  |  | |      | | | | ||
> > > |    SelfCheck             |84.32|84.9|85.03|86.39|15.89|94.26|13.78|95.17|
> > > |  LLMInertia               |89.31|85.4|88.03|89.14|11.22|95.56|10.47|96.10|
> > >
> > >
> > > **2.LLM-based Evaluation.** We deeply appreciate your follow-up suggestion and take your concern very seriously. As recommended, we conducted a comprehensive LLM-based evaluation using GPT-5. The results (see table below) indicate that LLMInertia exhibits superior performance in addressing unfaithful hallucinations. In addition, LLM-based and AlignScore evaluations exhibit marked Spearman correlations, further demonstrating the reliability of our evaluation.
> > >
> > >
> > > | Method | CNN/Daily Mail | SAMSum | SQuAD v2 | HaluEval |
> > > |-------------------------------------- |---------------|--------|----------|----------|
> > > |**LLaMA-3-Instruct-8B** | ||||
> > > | Base| 4.11| 4.18 | 4.6| 4.62 |
> > > | Prompt| 4.26| 4.25 | 4.63 | 4.57 |
> > > | CoT | 4.18| 4.24 | 4.62 | 4.69 |
> > > | SymbCoT | 4.21| 4.21 | 4.58 | 4.64 |
> > > | Lookback| 4.2 | 4.17 | 4.51 | **4.7**|
> > > | SFT | 4.32| 4.2| 4.51 | 4.68 |
> > > | **LLMInertia**| **4.54**| **4.58** | **4.79** | 4.68 |
> > > | *Spearman correlation with AlignScore*| *0.714* | *0.607*| *0.703*| *0.736*|
> > > | **LLaMA-3-Instruct-70B**| ||||
> > > | Base| 4.2 | 4.26 | 4.64 | 4.64 |
> > > | Prompt| 4.12| 4.3| 4.63 | 4.66 |
> > > | CoT | 4.31| 4.33 | 4.62 | 4.63 |
> > > | SymbCoT | 4.19| 4.4| 4.67 | 4.69 |
> > > | Lookback| 4.28| 4.28 | 4.59 | 4.6|
> > > | SFT | 4.39| 4.24 | 4.68 | 4.64 |
> > > | **LLMInertia**| **4.65**| **4.67** | **4.77** | **4.73** |
> > > | *Spearman correlation with AlignScore*| *0.75*| *0.685*| *0.75* | *0.721*|
> > > | **Qwen-2.5-Instruct-7B**| ||||
> > > | Base| 4.22| 4.11 | 4.32 | 4.52 |
> > > | Prompt| 4.33| 4.1| 4.37 | 4.54 |
> > > | CoT | 4.13| 4.16 | 4.27 | 4.6|
> > > | SymbCoT | 4.11| 4.18 | 4.33 | 4.59 |
> > > | Lookback| 4.15| 4.15 | 4.29 | 4.52 |
> > > | SFT | 4.16| 4.24 | 4.38 | **4.65** |
> > > | **LLMInertia**| **4.42** | **4.46** | **4.52** | 4.63 |
> > > | *Spearman correlation with AlignScore*| *0.643* | *0.679*| *0.685*| *0.649*|
> > >
> > > Overall, we appreciate your emphasis on the importance of LLM-based evaluations for a more comprehensive assessment. Although these evaluations can be costly, we have included them as far as our resources permit and will present these highly valuable results in the revised paper. Finally, we sincerely thank you for your thoughtful concerns, which have prompted us to conduct a more comprehensive and reliable evaluation that has ultimately strengthened our work.
> > >
> > > **3.Related Work.** We thank the reviewer for highlighting these important foundational works on faithfulness in abstractive summarization. While prior studies have established a strong foundation for addressing unfaithfulness in language models, our work focuses on the mechanisms underlying input-unfaithful hallucinations in LLMs. We are the first to systematically quantify how strong co-occurrence associations in LLMs trigger resistance to input evidence—a phenomenon we term "cognitive inertia." To address this, we propose adaptive counter-inertial reasoning to promote evidence-based and more reliable outputs. We will update the related work section in the revised paper to discuss and cite these works and to better positioning of our approach.
> > >
> > >
> > > Finally, we are truly grateful for your insightful suggestions and thoughtful concerns, which have led to valuable discussions and further enhanced the clarity and quality of our work. We will incorporate these discussions, along with the additional experimental analyses, into our revised manuscript. Once again, we thank you for your invaluable feedback.

---

### Official Review · Reviewer_1Kbd · 2026-03-07

**Soundness:** 3
**Presentation:** 3
**Significance:** 3
**Originality:** 3
**Overall Recommendation:** 5
**Confidence:** 4

**Summary:**

This paper studies input-evidence-unfaithful hallucinations in large language models (LLMs) and attributes them to a phenomenon termed cognitive inertia, where models over-rely on strong co-occurrence associations learned during pretraining and fail to adapt to contradictory contextual evidence. It explores the aspect of how strong co-occurrence associations in parametric memory affect model faithfulness to input evidence. Through controlled experiments manipulating co-occurrence frequency and training exposure, they show that stronger co-occurrence learning increases hallucination rates and reduces adherence to input evidence. To mitigate this issue, the paper proposes LLMInertia, a prompting framework that probes potential co-occurring associations and injects adaptive counter-inertial reminders into prompts to encourage evidence-based reasoning.

**Compliance With Llm Reviewing Policy:**

Affirmed.

**Key Questions For Authors:**

Can you briefly discuss the relationship between cognitive inertia and related concepts including as knowledge conflict, confirmation bias, and shortcut learning?

**Limitations:**

Yes

**Strengths And Weaknesses:**

Strengths:

The paper addresses an important and widely recognized challenge in LLM reliability, namely the tendency of models to ignore explicit contextual evidence when it conflicts with parametric knowledge. Understanding the role of co-occurrence bias in hallucinations contributes to the broader effort to improve trustworthy LLM behavior.

The paper presents a technically sound empirical investigation into how strong co-occurrence associations influence LLM faithfulness to input evidence. The controlled induction experiments that vary both co-occurrence data proportion and training epochs provide a clear experimental design to isolate the phenomenon of interest. The use of both synthetic entity pairs and medical pairs is a thoughtful design choice to distinguish newly induced biases from potential pre-existing associations in LLMs.

The work provides new empirical insight into the relationship between co-occurrence strength and hallucination behavior, which helps deepen understanding of how parametric knowledge interacts with contextual evidence.

The proposed hallucination mitigation framework is training-free and it can outperform SFT method in most of the evaluation metrics and datasets, which are impressive.

Weakness:

The notion of cognitive inertia is introduced as the central concept of the paper. However, its relationship to existing concepts such as knowledge conflict, confirmation bias, and shortcut learning is not sufficiently clarified. A clearer conceptual comparison with these existing ideas would strengthen the contribution.

Minor comments:

The authors could include additional error analysis to better understand the potential failure modes or limitations of the proposed LLMInertia prompting strategy.

Some relevant prior work on input-evidence-unfaithful hallucinations is missing in the related work section. For example:
1. Entity-based knowledge conflicts in question answering (ACL 2021)
2. Can LMs Generalize to Future Data? An Empirical Analysis on Text Summarization (EMNLP 2023)
3. Large Language Models Do NOT Really Know What They Don’t Know (Arxiv 2025)
These works analyze entity-level knowledge conflicts and hallucination mechanisms in QA and summarization tasks, which are closely related to the phenomenon studied in this paper.

---

> ### Author Rebuttal · Authors · 2026-03-30
>
> We deeply appreciate your thoughtful comments and your recognition of our work. Our detailed responses to your questions are provided.
>
> **1.Conceptual Clarification:** Thank you for raising this important question. Clarifying how cognitive inertia relates to similar concepts is essential for positioning our contribution. Below are brief definitions from the literature:
>
> - **Shortcut Learning:** Refers to  large language models to rely heavily on spurious correlations and non-generalized shortcuts in the training data rather than learning robust features for prediction [1].
> - **Knowledge Conflict:** Refers to situations where the evidence provided in the input is inconsistent with the model's memorized parametric knowledge [2].
> - **Confirmation Bias:** Describes LLMs demonstrate a strong confirmation bias when the external evidence contains some information that is consistent with their parametric memory, despite being presented with conflicting evidence at the same time [3].
>
> These concepts have drawn attention to and triggered reflection on unfaithful hallucinations in LLMs. Specifically, shortcut learning serves as the root cause: models internalize parametric priors from surface-level statistical correlations during pretraining. Furthermore, Knowledge conflict arises when input evidence contradicts these priors. In particular situations, confirmation bias describes the model's tendency to prefer memorized knowledge, even in the presence of conflicting information.
>
> Building on these insights, our work introduces the notion of cognitive inertia to help quantitatively connect these underlying causes with observable behaviors. Specifically, we provide the first systematic investigation into how excessive co-occurrence association within parametric priors triggers input-evidence-unfaithful hallucinations in LLMs, and we quantify their influence on the model’s resistance to explicit evidence. Furthermore, revealing this mechanism motivates the design of proactive interventions—adaptive counter-inertial reasoning—to address inertia-driven errors and promote evidence-grounded output.
>
> We will include this discussion in the revised paper.
> > [1] Robert Geirhos, et al. Shortcut learning in deep neural networks. Nat. Mach. Intell., 2(11):665–673, 2020.
> > [2] Hung-Ting Chen, et al. 2022. Rich knowledge sources bring complex knowledge conflicts: Recalibrating models to reflect conflicting evidence. ArXiv preprint, abs/2210.13701.
> > [3] Xie, J, et al. Adaptive chameleon or stubborn sloth: Revealing the behavior of large language models in knowledge conflicts. ICLR, 2023.
>
> **2. Supplementary Error Analysis:** Thank you for the question. Below, we summarize key observations from our error analysis:
>
> - **Probing Failures:** The probing stage can sometimes fail to accurately capture the model’s inertia entities, either by missing the most over-learned associations or by error propagation from incomplete or imprecise entity extraction, resulting in missed key reminders or distracting ones.
> - **Stubborn Cognitive Inertia:** For some deeply embedded co-occurrence associations, even accurate adaptive reminders may be ignored, and hallucinations persist.
> - **Oscillating in Complex Cases:** In certain complex semantic or reasoning scenarios, the model may initially respond to the reminder but later revert during inference, oscillating and ultimately producing unfaithful answers.
>
> We will include this discussion in the revised paper.
>
> **3.Discussion of Related Work:** Thank you for this valuable suggestion. We fully agree that these studies are closely relevant and will be cited and discussed in the revised manuscript.
>
> - **Entity-based Knowledge Conflicts in QA:** formalizes knowledge conflicts where contextual evidence contradicts memorized parametric knowledge, and analyzes the over-reliance on memorized information as a cause of hallucination.
> - **Can LMs Generalize to Future Data?:** introduces the TempoSum benchmark to understand the temporal generalization ability of abstractive summarization models. The results show that parametric knowledge stored in summarization models significantly affects the faithfulness of generated summaries on future data.
> - **LLMs Do NOT Really Know What They Don’t Know:** finds that when hallucinations are produced from learned associations, their hidden-state geometries largely overlap with factual outputs, rendering standard detection methods ineffective.
>
> While these works provide valuable insights, they do not systematically quantify how the strength of co-occurrence associations within parametric priors impacts a model’s resistance to explicit input evidence. Our work is the first to systematically quantify this effect—termed cognitive inertia—demonstrating that it intensifies with increased co-occurrence frequency and training duration, and motivating adaptive counter-inertial reasoning to promote evidence-grounded output.

---

> > ### Author Rebuttal · Reviewer_1Kbd · 2026-04-03
> >
> > The authors clarified  how cognitive inertia relates to similar concepts and discussed error analysis observations

---

> > > ### Author Response · Authors · 2026-04-08
> > >
> > > We greatly appreciate your support and positive feedback, and are delighted to learn that our rebuttal has fully addressed your concerns. We sincerely thank you for your time, effort, and insightful comments throughout the review process. Your feedback has provided invaluable guidance in improving our work.

---

### Official Review · Reviewer_QLmd · 2026-03-08

**Soundness:** 3
**Presentation:** 3
**Significance:** 3
**Originality:** 3
**Overall Recommendation:** 5
**Confidence:** 4

**Summary:**

The authors show that LLMs can be subject to cognitive inertia, where correlated facts seen during pre-training can result in ignoring input evidence downstream.

They also propose a novel method for mitigating this issue.

**Compliance With Llm Reviewing Policy:**

Affirmed.

**Key Questions For Authors:**

- What's the compute overhead of using your method?
- There are plenty of ways one could get an LLM to focus on the provided evidence. What motivates your specific implementation?
- Do you perform instruction-tuning after continually pre-training these models?
- Who should use your method and when?
- In experiments on QA tasks, you report Anah-v2 and AlignS scores, but how is accuracy impacted?
- Did you fine-tune the prompts used in the paper?
- Have you tried some generic prompt that gets the model to do the same careful examination, without needing to extract specific correlated entities?

**Limitations:**

I think that limitations could be more thoroughly discussed. For instance, it seems like the proposed method could potentially have issues with scaling to large texts. Additionally, it is not clear if there are downsides to the method, such as cases where it is detrimental to use it.

**Strengths And Weaknesses:**

# Strengths
- Clear problem statement
- Nice, simple idea
- Experiments back the authors' claim
- Well written

# Weaknesses
- Continual pre-training can be tricky, especially when data is limited. Have you looked at how the overall quality of the models diminishes as you continually pre-trained these models, potentially due to overfitting? Have you taken this into consideration, and how do you disentangle this factor from the observed performance trends?
- For very long texts, wouldn't entity extraction become a problem?

---

> ### Author Rebuttal · Authors · 2026-03-30
>
> We sincerely appreciate your feedback and recognition of our work. Below, we address each of the concerns you raised.
>
> **1. Discussion on Induction Experiments:** To prevent overfitting, epoch-varying training sets were constructed with 25% induction data and 75% Wiki-en text (see page 4, line 245), ensuring reasonable size and diversity.
> We also assessed LLaMA-3-8B on GSM8K and MMLU under 5-shot settings before and after 10 epochs of medical induction training, observing that it preserved over 95% of its initial accuracy (GSM8K: 73.2%→70%; MMLU: 64.5%→61.5%). Based on these results, a rough estimate suggests that of the 39% reduction in accuracy after 10 epochs, only about 2% can be attributed to overfitting, while the remaining 37% is due to accumulated cognitive inertia.
>
> **2.Scalability in Long Texts:** To assess scalability, we benchmarked LLMInertia’s overhead as the number of extracted entities increased on the CNN/Daily Mail dataset:
>
> |#Entities|Tokens|Latency(s)|
> |-|-|-|
> |8|2286|2.1|
> |16|2383|2.6|
> |24|2481|3.1|
> |32|2579|3.7|
>
> Even quadrupling the number of entities led to acceptable increases in token and latency. This efficiency due to design choices that mitigate slower decoding: prompt design for concise outputs (Appendix C.1), compact reminders (avg. 3.9 for CNN/Daily Mail; see Figure 5), and batching entities in one probe request for efficiency.
>
> **3.Overhead of LLMInertia:** We measured token usage and latency (NVIDIA A100-80GB with vLLM).
>
> **Table: Average token and latency overhead (tokens / seconds). The last two columns indicate the fold increase over baseline.**
>
> |Task/Dataset|Base|Prompt| LLMInertia|LLMInertia vs. Base|LLMInertia vs. Prompt|
> |-|-|-|-|-|-|
> |Summary / CNN/Daily Mail|1105 / 1.23|1134 / 1.29| 2476 / 3.31| 2.2x / 2.7x| 2.2x / 2.6x |
> |QA / SQuAD v2|272 / 0.23| 298 / 0.28|711 / 0.6|2.6x / 2.6x| 2.4x / 2.1x|
>
> The main source of overhead is the increased input from entity extraction, which requires full context. However, since LLMs process input efficiently, this overhead is relatively minor and can be further reduced in practice via external entity extraction or caching frequently co-occurring entity pairs.
>
> **4.Design Motivation for LLMInertia:** Our implementation is motivated by our systematic investigation of cognitive inertia in LLMs—the tendency to rely on co-occurrence associations from pretraining even when faced with conflicting input evidence—which our experiments show to be a significant cause of input-unfaithful hallucinations.
>
> Building on this insight and inspired by human counter-inertial reasoning—where people identify counter-intuitive information and emphasize critical cues to ensure faithful decisions—our method mirrors this process in three steps:
> - 1) probing the LLM to uncover input-relevant inertial associations;
> - 2) generating adaptive reminders that highlight pitfalls in the input;
> - 3) injecting these reminders into the prompt, guiding the model’s focus toward evidence-based reasoning.
>
> This targeted approach effectively mitigates cognitive inertia and produces more faithful outputs.
>
> **5.Supplementary QA Results.** We additionally evaluated QA tasks with LLaMA-3-Instruct-8B using F1 and Exact Match (EM). As shown below, LLMInertia achieves the highest scores, indicating that mitigating cognitive inertia improves both faithfulness and QA quality.
> |Method|SQuAD V2 (F1)|SQuAD V2 (EM)|HaluEval (F1)|HaluEval (EM)|
> |-|-|-|-|-|
> |Base|83.59|79.07|81.33|77.46 |
> |Prompt| 84.37|80.21|84.69|80.52|
> |CoT| 82.09| 86.14|85.37|81.64|
> |SymbCoT| 82.14|86.20| 85.44|81.73|
> |Lookback| 81.63|80.81| 83.69| 82.25|
> |SFT|82.27| 84.24|85.63| 81.14|
> |LLMInertia| 85.92|87.54| 87.10 |84.78|
> *All values are percentages*
>
> **6. Generic Prompt Experimentation:** We additionally evaluated a generic prompt (line 272, footnote). As shown below, LLMInertia outperforms it by 0.71%–2.79% across all metrics, further confirming LLMInertia’s effectiveness.
>
> |Methods|CNN/Daily Mail (Consistency↑)|CNN/Daily Mail (AlignScore↑)|SQuAD v2(Anah-v2↓)|SQuAD v2 (AlignScore↑)|
> |-|-|-|-|-|
> |Generic Prompt|87.54(-2.79)|86.58 (-1.66)|14.98 (-2.45) |95.63 (-0.71) |
> |LLMInertia|90.33|88.24|12.53|96.34|
>
> **7.Implementation Details, Suitability, and Limitations:**
>
> *Instruction Tuning:* We did not perform instruction tuning or SFT after continual pre-training to isolate the effect of cognitive inertia and ensure a clear assessment of faithfulness changes.
>
> *Prompt Design:* All prompts were manually crafted and refined for clarity; no automated or gradient-based prompt tuning was used.
>
> *Suitability:* LLMInertia is task-agnostic and training-free, making it suitable for practitioners, developers, and researchers needing reliable outputs and high-stakes scenarios demanding strict evidence faithfulness.
>
> *Limitations:* We will thoroughly discuss limitations, including computational overhead on long texts. No other downsides have been identified so far, but we will continue to monitor for limitations.

---

> > ### Author Rebuttal · Reviewer_QLmd · 2026-04-06
> >
> > Thank you very much for your rebuttal!

---

> > > ### Author Response · Authors · 2026-04-08
> > >
> > > We are truly grateful for your positive feedback and encouragement, and for letting us know that our rebuttal has fully addressed your concerns. We deeply appreciate the time and thoughtful consideration you devoted to our submission. Your constructive comments have greatly contributed to improving our work !

---

### Official Review · Reviewer_p2RK · 2026-03-12

**Soundness:** 3
**Presentation:** 3
**Significance:** 3
**Originality:** 3
**Overall Recommendation:** 4
**Confidence:** 3

**Summary:**

This paper investigates the phenomenon of cognitive inertia in LLMs—a tendency to overly rely on co-occurrence associations learned during pretraining and to resist adaptation when conflicting evidence appears in the input. The authors aim to explore the aspect of input-evidence-unfaithful hallucinations through two complementary angles: (1) a preliminary empirical investigation demonstrating that strengthening co-occurrence associations—via increased data frequency or prolonged training—exacerbates hallucinations, and (2) a training-free mitigation framework, LLMInertia, which probes input-relevant inertial associations within the LLM, generates adaptive counter-inertial reminders, and injects them into the prompt to promote evidence-faithful reasoning. The work addresses a major problem in the reliable deployment of LLMs, particularly in high-stakes domains such as healthcare and law. Experiments are conducted across three backbones (LLaMA-3-8B, Qwen-2.5-7B, LLaMA-3-70B), two co-occurrence induction paradigms, and four downstream summarization and QA benchmarks.

**Compliance With Llm Reviewing Policy:**

Affirmed.

**Final Justification:**

The authors' responses have addressed most of my concerns. Based on the new findings during the discussion phase, I decide to raise the evaluation to weak accept.

**Key Questions For Authors:**

Q1. Regarding the epoch-varying experiments in Figure 3: how can the authors rule out that the severe performance degradation observed at 10 training epochs is attributable to overfitting on the small, homogeneous induction corpus, rather than to the genuine accumulation of co-occurrence-driven cognitive inertia? Could the authors provide evidence—such as perplexity on held-out general text, general instruction-following benchmarks, or comparison against a control group trained on random data of the same size and duration—to disentangle these concerns?

Q2. Could the authors provide a systematic ablation study isolating the contribution of each component of LLMInertia (entity extraction, co-occurrence probing, adaptive reminder generation, and reminder injection)?

Q3. What is the average token overhead and wall-clock latency of LLMInertia relative to the Base and Naive Prompting baselines, across the evaluated benchmarks? How does this overhead scale with the number of entities extracted from the input?

**Limitations:**

yes

**Strengths And Weaknesses:**

**Strengths**
1.  The authors identify a clear and practically significant target: input-evidence-unfaithful hallucinations driven by over-learned co-occurrence associations.

2. LLMInertia is elegantly designed, which leverages the model's own parametric priors to identify high-frequency co-occurrence associations.

3. The authors evaluate LLMInertia across multiple LLM backbones, two co-occurrence induction paradigms (data proportion and training epoch manipulation), and four context-rich NLP benchmarks, providing empirical breadth and analytical depth.

**Weeknesses**
1. The paper does not provide ablation experiments that systematically isolate the contribution of each component. For instance, it is unclear whether the performance gains are primarily attributable to the adaptive, entity-specific reminders, or whether a simpler alternative (such as injecting the generic reminder uniformly for comparison).

2. The LLMInertia pipeline requires three sequential LLM calls per input (entity extraction, co-occurrence probing, and final inference with injected reminders), with the number of probing calls scaling with the number of extracted entities. The main paper does not report token overhead and latency relative to the baselines.

---

> ### Author Rebuttal · Authors · 2026-03-30
>
> We sincerely appreciate your feedback and recognition of our work. Below are our responses to each of the concerns you raised.
>
> **1. Ablation Study of LLMInertia:** Thank you for your suggesting feedback. We conducted a systematic ablation experiments with LLaMA-3-Instruct-8B on CNN/Daily Mail (summarization) and SQuAD v2 (QA). For each ablation, one key component was removed with other settings kept as unchanged as possible.
>
> - **w/o Entity Extraction:** Entities are not extracted; co-occurrence probing and reminder cannot be performed. Thus, a generic reminder (see line 272, footnote) is injected into all samples.
> - **w/o Inertia Probe:** Entities are extracted, but high-risk pairs are not identified via LLM probing. Instead, a sliding window (size=5) determines co-occurrence pairs. (Exhaustive pairing was overly noisy; sliding window balances converage and performance.)
> - **w/o Adaptive Reminder:** Upon detecting high-risk pairs, only a fixed generic reminder is injected, not inertia-specific adaptation.
>
> **Table: Ablation results for LLMInertia (%).**
>
> |Methods|CNN/Daily Mail (Consistency↑) |CNN/Daily Mail (AlignScore↑) | SQuAD v2 (Anah-v2↓) | SQuAD v2 (AlignScore↑) |
> |-|-|--|--|--|
> | LLMInertia   | 90.33   | 88.24| 12.53| 96.34 |
> | w/o Entity Extraction| 87.54 (-2.79) | 86.58 (-1.66) | 14.98 (-2.45)  | 95.63 (-0.71)  |
> | w/o Inertia Probe| 85.76 (-4.57)  | 82.85 (-5.39) | 17.48 (-4.95)  | 92.15 (-4.19) |
> | w/o Adaptive Reminder| 87.03 (-3.30) | 86.44 (-1.80) | 14.99 (-2.46) | 95.24 (-1.1)|
>
> As shown, values in parentheses indicate the decrease for each metric relative to LLMInertia. The largest drop occurs without co-occurrence probing, underscoring its key role in identifying the LLM’s internal co-occurrence biases. Removing adaptive reminders leads to a moderate decline, indicating their necessity for focusing the model on error-prone evidence. Omitting entity extraction, which leads to general reminders, results in a slight reduction. Overall, these results demonstrate that all components are essential and complementary for improving evidence faithfulness.
>
> **2. Overhead and Scalability of LLMInertia:** We benchmarked LLMInertia on LLaMA-3-Instruct-8B (single NVIDIA A100-80GB with vLLM), measuring token usage and latency on representative datasets (due to time constraints), the results are shown in the table below. The main source of overhead in LLMInertia is the increased input from entity extraction, which requires processing the full input context. However, since LLMs process input efficiently, this overhead is relatively minor and can be further reduced in practice via external entity extraction or caching frequently co-occurring entity pairs. The primary bottleneck lies in decoding; to address this, we have optimized our approach through prompt design for concise outputs (see Appendix C.1) and by compact reminders (averaging 3.9 per CNN/Daily Mail sample; see Figure 5) to prevent unnecessary outputs.
>
> **Table: Average token and latency overhead (tokens / seconds). The last two columns show the times increase over the baselines.**
>
> |Task/Dataset|Base|Prompt| LLMInertia|LLMInertia vs. Base|LLMInertia vs. Prompt|
> |-|-|-|-|-|-|
> |Summary / CNN/Daily Mail|1105 / 1.23|1134 / 1.29| 2476 / 3.31| 2.2x / 2.7x| 2.2x / 2.6x |
> |QA / SQuAD v2|272 / 0.23| 298 / 0.28|711 / 0.6|2.6x / 2.6x| 2.4x / 2.1x|
> |QA / HaluEval|242 / 0.22| 268 / 0.27|658 / 0.57|2.7x / 2.6x| 2.5x / 2.1x|
>
>
> **For scalability**, we clarify that co-occurrence probing is performed by aggregating all entities into a single request (see Appendix C.1.2) to improve efficiency. We assessed scalability on the context-rich CNN/Daily Mail by prompting the LLM to extract only the top-N most important entities per input (N = 8, 16, 24, 32).
>
> |#Entities|Tokens|Latency(s)|
> |-|-|-|
> |8|2286|2.1|
> |16|2383|2.6|
> |24|2481|3.1|
> |32|2579|3.7|
>
> Increasing the number of extracted entities from 8 to 32 leads to acceptable growth in token usage and latency, reflecting cost control. We consider this moderate, optimizable overhead acceptable for practical applications, given the improvements in reliability.
>
> **3. Discussion on Induction Experiments:** We thank the reviewer for this insightful question. To minimize overfitting, our epoch-varying training sets included 25% co-occurrence induction data and 75% Wiki-en text (see page 4, line 245), ensuring appropriate size and diversity.
>
> We further evaluated LLaMA-3-8B on GSM8K and MMLU under 5-shot settings before and after 10 epochs of medical induction training, and found that it retained over 95% of its original accuracy (GSM8K: 73.2%→70%; MMLU: 64.5%→61.5%). Based on these results, a rough estimate suggests that of the 39% reduction in accuracy after 10 epochs, only about 2% can be attributed to overfitting, while the remaining 37% is due to accumulated cognitive inertia.

---

> > ### Author Rebuttal · Reviewer_p2RK · 2026-04-03
> >
> > Thank the authors for the detailed rebuttal and the additional results provided. While the responses partially address some of my questions, several concerns remain insufficiently resolved:
> >
> > 1) The reported "2.2–2.7× overhead" is non-trivial and may be a concern in latency-sensitive scenarios. It would be helpful to examine how this overhead scales with input context length, which is arguably more relevant in practice.
> >
> > 2) The estimate that only "~2% of the degradation is due to overfitting" relies on a rough calculation from two randomly selected benchmarks. It may not be sufficient to draw a definitive conclusion until more detailed comparative analysis. A more controlled experiment, such as training on non-co-occurrence data of the same size, would more convincingly disentangle overfitting from the proposed cognitive inertia effect.

---

> > > ### Author Response · Authors · 2026-04-06
> > >
> > > We greatly appreciate the time and effort you have invested in reviewing our submission. Your ongoing feedback has been instrumental in enhancing the quality of our manuscript. We truly value your concerns and are dedicated to responding to them thoughtfully.
> > >
> > > **1. Scalability of LLMInertia:** Thank you very much for your valuable suggestion regarding the scaling of overhead with input context length. Following your advice, we performed additional empirical evaluation on LLMInertia's latency using input texts of varying lengths (500, 1000, 2000 from CNN/Daily Mail; 4000 from LongBench[1]). The results are summarized in the table below:
> > >
> > > **Table: Average latency overhead (seconds). The last two columns show the times increase over the baselines.**
> > >
> > > | Input Length | Base Latency  | Prompt Latency | LLMInertia Latency | LLMInertia vs. Base | LLMInertia vs. Prompt |
> > > |--------------|--------------|----------------|--------------------|---------------------|----------------------|
> > > | 500      | 0.5      | 0.6        | 1.2        | 2.2x        | 2.1x                 |
> > > | 1000     | 1.2      | 1.3      | 3.1          | 2.58x       | 2.38x                 |
> > > | 2000    | 1.7      | 1.8      | 3.7     | 2.18x         | 2.05x                 |
> > > | 4000     | 2.6      | 2.9     | 5.6        | 2.15x     | 1.93x                 |
> > >
> > > We observe that LLMInertia’s latency overhead remains moderate as input length increases, with the relative gap to baselines first rising and then falling. For inputs of 500–1000 tokens, the low baseline latency makes pipeline scheduling comparatively costly, resulting in an increased overhead ratio. As input length grows (1000 → 2000 → 4000), the baseline latency increases accordingly, while LLMInertia’s cost increases more slowly—mainly depending on the number of extracted entities (e.g., the extraction step lists all important entities, and the probing step returns the most closely related entities; see Appendix C.1). Thus, the overhead is increasingly amortized over longer contexts, with the ratio dropping. These results demonstrate that LLMInertia’s overhead growth is manageable and supports efficient scalability for long-input scenarios.
> > >
> > > For latency-sensitive scenarios, optimizations such as external entity extraction or prefix caching can effectively reduce processing time. Given the reliability gains—especially for safety-critical domains(e.g., medical, legal)—we consider the moderate and optimizable overhead an acceptable tradeoff in practice.
> > >
> > > We sincerely appreciate your suggestion, which prompted this more comprehensive analysis, and we will include these significant discussions in the revised paper.
> > >
> > > > [1] Bai Y, Lv X, Zhang J, et al. Longbench: A bilingual, multitask benchmark for long context understanding[C]//Proceedings of the 62nd annual meeting of the association for computational linguistics (volume 1: Long papers). 2024: 3119-3137.
> > >
> > > **2.Discussion on Induction Experiments:** Thank you for your follow-up suggestion. We fully understand your concern. Following your advice, we conducted additional control experiments using only non-co-occurrence data (100% Wiki-en), with the same data size and training duration. The results are shown below:
> > >
> > > | Training Data                | Test data     | Accuracy: Before → After | Accuracy Drop |
> > > |------------------------------|------------|-------------------------|---------------|
> > > | 25% induction + 75% Wiki-en  | Synthetic  | 86.02% → 48.94%         | 37.08%        |
> > > | 100% Wiki-en                 | Synthetic  | 86.02% → 83.43%         | 2.59%         |
> > > | 25% induction + 75% Wiki-en  | Medical    | 89.82% → 24.11%         | 65.71%        |
> > > | 100% Wiki-en                 | Medical    | 89.82% → 86.23%         | 3.59%         |
> > >
> > > As shown, in the experimental group with 25% co-occurrence induction data, downstream accuracy dropped sharply, while the control group exhibited a much smaller accuracy decrease. Collectively, these results provide strong evidence that the performance degradation in Figure 3  is mainly caused by the accumulation of co-occurrence-driven cognitive inertia, rather than overfitting.
> > >
> > > Finally, we sincerely thank you for your insightful suggestions and thoughtful concerns, which have led to meaningful discussions and improved the clarity and rigor of our work. We will incorporate these discussions and additional experimental analyses in the revised manuscript. We again thank you for your valuable feedback.

---

### Decision · Program_Chairs · 2026-04-30

**Decision:**

Accept (regular)

**Comment:**

This paper studies an important failure mode in LLMs where models ignore explicit input evidence and instead follow strong co-occurrence associations learned during pretraining. The authors argue that this behavior can be understood as cognitive inertia and supports that claim with controlled induction experiments that vary both co-occurrence frequency and training exposure. It then proposes a simple inference-time mitigation called LLMInertia that probes likely inertial associations and injects targeted reminders to redirect reasoning back toward the input. The  authors present reasonably broad empirical support for both the phenomenon and the mitigation. In those evaluations, LLMInertia generally improves over prompt-based and other training-free baselines, and the manual analysis suggests that the targeted error type accounts for a meaningful subset of unfaithful outputs.

Concerns raised by the reviewers were mainly asking for stronger evidence that the induction results were not just artifacts of generic overfitting, a clearer ablations of the method’s components, stronger comparison to a simple self-check baseline, more comprehensive LLM-based evaluation, and better conceptual positioning relative to neighboring notions such as knowledge conflict, confirmation bias, shortcut learning, and earlier faithfulness work. The authors made a substantial effort in rebuttal and added component ablations and overhead measurements and input-length scaling results. They also provided a non-co-occurrence control condition that greatly reduces the degradation seen in the induction setting and added GPT-5-based evaluation that manly agrees with the original automatic metrics.  As some of the strongest evidence for the work was generated during the rebuttal, it is very important for those to be reflected and integrated into the paper itself.

Even with these new experiments, I still however see a few limitations. The authors should be careful not to overstate the scope of the contribution. Even by the authors' own manual analysis, cognitive inertia appears to explain about one third of unfaithful cases on the examined datasets, which is important but not exhaustive. Also, the method has nontrivial inference overhead, on the order of roughly two to three times increase in the reported experiments, which seems acceptable in reliability-sensitive settings but is still a real practical cost. Finally, the related-work framing in the submission appears thinner than it should be, especially around prior faithfulness work in summarization and adjacent conceptual framing.

I really do not think the conceptual novelty should be overstated. But the paper tells a coherent empirical story otherwise and the authors are addressing an important problem and provide a simple mitigation strategy.